TOPICAL REVIEW

# The diverse functions of the DEG/ENaC family: linking genetic and physiological insights

Eva Kaulich[1] , Laura J. Grundy[1], William R. Schafer[1,2]  and Denise S. Walker[1] 

[1]*Neurobiology Division, MRC Laboratory of Molecular Biology, Francis Crick Avenue, Cambridge Biomedical Campus, Cambridge, UK*
[2]*Department of Biology, KU Leuven, Leuven, Belgium*

Handling Editors: Laura Bennet & Morag Mansley

The peer review history is available in the Supporting Information section of this article (https://doi.org/10.1113/JP283335#support-information-section).

**Abstract** The DEG/ENaC family of ion channels was defined based on the sequence similarity between degenerins (DEG) from the nematode *Caenorhabditis elegans* and subunits of the mammalian epithelial sodium channel (ENaC), and also includes a diverse array

 

**Eva Kaulich** and **Laura Grundy** are Postdoctoral Researchers, and **Denise Walker** is a Senior Scientist, in **William Schafer's** laboratory at the MRC Laboratory of Molecular Biology. They share a fascination for ion channels and bridging the gap between genetics, physiology and behaviour, using the nematode *C. elegans*. The lab focuses on how ion channels and receptors contribute to nematode behaviour and has carried out large-scale screens to characterize the expanded families of nematode ion channels, using electrophysiological techniques, in order to investigate how they function *in vivo*.

of non-voltage-gated cation channels from across animal phyla, including the mammalian acid-sensing ion channels (ASICs) and *Drosophila* pickpockets. ENaCs and ASICs have wide ranging medical importance; for example, ENaCs play an important role in respiratory and renal function, and ASICs in ischaemia and inflammatory pain, as well as being implicated in memory and learning. Electrophysiological approaches, both *in vitro* and *in vivo*, have played an essential role in establishing the physiological properties of this diverse family, identifying an array of modulators and implicating them in an extensive range of cellular functions, including mechanosensation, acid sensation and synaptic modulation. Likewise, genetic studies in both invertebrates and vertebrates have played an important role in linking our understanding of channel properties to function at the cellular and whole animal/behavioural level. Drawing together genetic and physiological evidence is essential to furthering our understanding of the precise cellular roles of DEG/ENaC channels, with the diversity among family members allowing comparative physiological studies to dissect the molecular basis of these diverse functions.

(Received 6 July 2022; accepted after revision 25 October 2022; first published online 4 November 2022)
**Corresponding author** D. S. Walker: Neurobiology Division, MRC Laboratory of Molecular Biology, Francis Crick Avenue, Cambridge Biomedical Campus, Cambridge CB2 0QH, UK.　　Email: dwalker@mrc-lmb.cam.ac.uk

**Abstract figure legend** Schematic illustration of the diverse modulators and stimuli that influence DEG/ENaC function (blue arrows) and the diverse range of functions in which they have been implicated (green arrows), in an array of animals, exemplified by those depicted. Created with Biorender.com.

## Introduction: forward genetics meets channel physiology, the founding of a superfamily

The founding of the DEG/ENaC family arose from the convergence of electrophysiological and genetic evidence. Mutations in so-called degenerin genes *mec-4* and *mec-10* were found in forward genetic screens, i.e. screens that begin with the identification of a phenotype (as opposed to reverse genetics, where the starting point is a candidate gene(s)). Chemically mutated *Caenorhabditis elegans* were screened for touch insensitive mutants (Mec, mechanosensation abnormal phenotype) (Chalfie & Sulston, 1981; Waldmann et al. 1996), some of which exhibited degeneration of the touch-sensing 'touch receptor neurons' (TRNs). Mutations in another gene, *deg-1*, caused degeneration of a different set of neurons. Cloning of the *deg-1*, *mec-4* and *mec-10* genes revealed that they encoded homologous integral membrane proteins that were putative channel subunits (Chalfie & Wolinsky, 1990; Chalfie et al. 1993; Driscoll & Chalfie, 1991; Huang & Chalfie, 1994). Meanwhile, an expression cloning approach, in which rat cDNAs were screened for amiloride-sensitive ENaC channel properties in *Xenopus* oocytes (Canessa et al. 1993, 1994; Lingueglia et al. 1993) identified the genes encoding ENaC subunits, which shared homology with the degenerins. Further genetic screens and expression cloning approaches across various phyla identified other members of this DEG/ENaC family with diverse physiological properties and functional roles. With genome sequencing, it became clear that both *C. elegans* and *Drosophila* have vastly expanded DEG/ENaC families compared to mammals.

Here, we provide an overview of the evidence from diverse experimental approaches, encompassing physiological characterisation, genetics and behaviour, from examples spanning vertebrate and invertebrate phyla. They reveal a diversity of channel properties, with a wide range of ligands and modulators, and a wide range of functions in diverse cellular contexts. However, our knowledge is far from complete; the extent to which properties are conserved between family members and across phyla is unclear, requiring more extensive characterisation. We discuss how learning from studies on other family members and drawing together structural, physiological, genetic and behavioural evidence will be important in the quest to better understand the molecular and cellular basis of their function and dysfunction.

## Structural insights into DEG/ENaC function

Until recently, structural knowledge of DEG/ENaCs was primarily dependent on the X-ray crystal structure of a homomeric chicken acid-sensing ion channel (ASIC), composed of a truncated version of ASIC1 (Jasti et al. 2007; Yoder & Gouaux, 2018), with subsequent electron cryo-microscopy (cryoEM) structures of truncated human ENaC expressed as $\alpha\beta\gamma$ heteromers (Noreng et al. 2018, 2020) and full length chicken ASIC1 (Yoder & Gouaux, 2020) adding further mechanistic insight. Both form a trimer, with individual subunits displaying a similar domain organisation: a large globular extracellular domain

(ECD), with multiple subdomains and a transmembrane domain consisting of two transmembrane helices.

The structure of chicken ASIC1 (Jasti et al. 2007; Yoder & Gouaux, 2018, 2020) resembles a hand holding a ball (Fig. 1). Of the ECD subdomains, the 'palm' is formed by the twisted $\beta$-strand-rich $\beta$-sheet, the 'thumb' and 'finger' are formed by outer $\alpha$-helix-rich domains, and the central $\beta$-strand-rich region forms the 'ball'. The interface between the thumb, $\beta$-ball and finger region encompasses the primary acidic pocket, with additional contributions from the palm subdomain from the adjacent monomer (Fig. 1*A* and *B*). There are several proposed proton binding sites, one of which is this acidic pocket, which is border-lined by negatively charged residues, thereby contributing to channel gating properties (Jasti et al. 2007; Vullo et al. 2017). However, there are other proposed sites outside the acidic pocket (Chen et al. 2021; Ishikita, 2011; Li et al. 2010; Rook et al. 2021). These structures, together with functional studies, identified the existence of an upper gate in ASIC1, contributed by Asp433 (numbering in chicken ASIC1) from each monomer (Fig. 1*E*). Two highly conserved negatively charged residues, Glu451 and Asp454, have been reported to be crucial contributors to the selectivity filter, in both ASICs and ENaCs (Lynagh et al. 2017; Yang & Palmer, 2018), although a recent study suggested that their effect is indirect, via interactions with the 'GAS' belt (described below) (Chen et al. 2022). The recent structures of full-length chicken ASIC1 (Fig. 1*C* and *D*), determined by cryoEM, additionally revealed an N-terminal re-entrant loop (Re-1 and 2), with a highly conserved 'His–Gly' (HG) motif. Even more interestingly, this HG motif is the residence for a second, lower, channel gate, formed by three histidines (His29) Fig. 1*F*. The HG motif is in close proximity to another conserved and critical stretch of 'Gly–Ala–Ser' residues which forms a 'GAS' belt (Yoder & Gouaux, 2020) (Fig. 1*G*). Both motifs are highly conserved across DEG/ENaC family members and have been implicated in gating and ion selectivity (Chen et al. 2022; Grunder et al. 1999; Kellenberger & Schild, 2002; Kellenberger et al. 1999a, 1999b, 2001, 2003).

The DEG/ENaC channels show a diversity in ion selectivity and permeability; while many are Na$^+$-selective, some are also permeable to K$^+$ or Ca$^{2+}$, and some invertebrate members exhibit a greater relative selectivity for other cations than Na$^+$ (Dürrnagel et al. 2012; Fechner et al. 2021; Kaulich et al. 2022b; Wang et al. 2008; further reviewed in Vallee et al. 2021).

## Invertebrate models for DEG/ENaC roles in mechanosensation

Since the discovery of MEC-4 and MEC-10, worms have proven useful models for informing our understanding of mechanosensation. *C. elegans* responds with readily assayable behaviours that can be harnessed as a readout of cellular function. The predominant response to a noxious mechanical stimulus or obstruction is escape, a rapid change in locomotion direction, as in the gentle touch assay (Fig. 2*A*). *C. elegans* exhibits other mechanosensory behaviours, such as slowing on food and proprioceptive modulation of locomotion (Goodman & Sengupta, 2019; Hesselson et al. 2020; Schafer, 2015).

*mec-4* is expressed exclusively in the TRNs (Fig. 2*B*) and is essential for the escape response to gentle touch (Huang & Chalfie, 1994) and mechanically induced currents (O'Hagan et al. 2005) and Ca$^{2+}$ transients in the TRNs, but dispensable for harsh touch responses (Suzuki et al. 2003) (Fig. 2*C*). MEC-10 is also expressed in the TRNs, functioning in both gentle and harsh touch (Arnadottir et al. 2011; Chatzigeorgiou et al. 2010a). It also plays multiple roles in two pairs of intricately branched multifunctional nociceptors, called FLP and PVD. In the FLPs it functions in the responses to nose touch and harsh head touch (Chatzigeorgiou & Schafer, 2011). In the PVDs it functions along with two other DEG/ENaC subunits, DEL-1 (named for its homology, DEgenenerin Like) and UNC-8 (named for the mutant phenotype, UNCoordinated), in proprioception (Albeg et al. 2011; Tao et al. 2019); its role in PVD in harsh touch is more controversial, with conflicting evidence that it is required or dispensable (Chatzigeorgiou et al. 2010b; Li et al. 2011a; Tao et al. 2019). The explanation may lie in differences in the behavioural and functional assays used. It is nevertheless clear that, within a given neuron class, different DEG/ENaC channels perform distinct roles, detecting different mechanical stimuli and initiating distinct downstream effects. In PVD, for example, proprioception initiates local dendritic Ca$^{2+}$ increases and neuropeptide release, locally modulating neuromuscular junction activity (Tao et al. 2019), whereas nociception triggers a global Ca$^{2+}$ transient and activation of the escape circuit.

MEC-4 can form homotrimers or 2:1 heteromers with MEC-10, whereas MEC-10 depends on MEC-4 to form functional channels (Chen et al. 2015, 2016; O'Hagan et al. 2005). Thus, in the TRNs, an attractive hypothesis is that heteromers detect higher threshold and MEC-4 homomers detect lower threshold stimuli, in agreement with the genetic evidence for their distinct roles outlined above. Demonstrating mechanosensitivity of heterologously expressed channels has, however, proved challenging. A response to sheer force has been demonstrated for MEC-4–MEC-10 heteromers (Shi et al. 2016), the same stimulus used for ENaC (Carattino et al. 2004), demonstrating the importance of learning from approaches used to characterize family members from other phyla. But the same approach was not successful for MEC-4 alone, which may require a different stimulation protocol or be more dependent on cellular context. Several other *C. elegans* members are implicated in

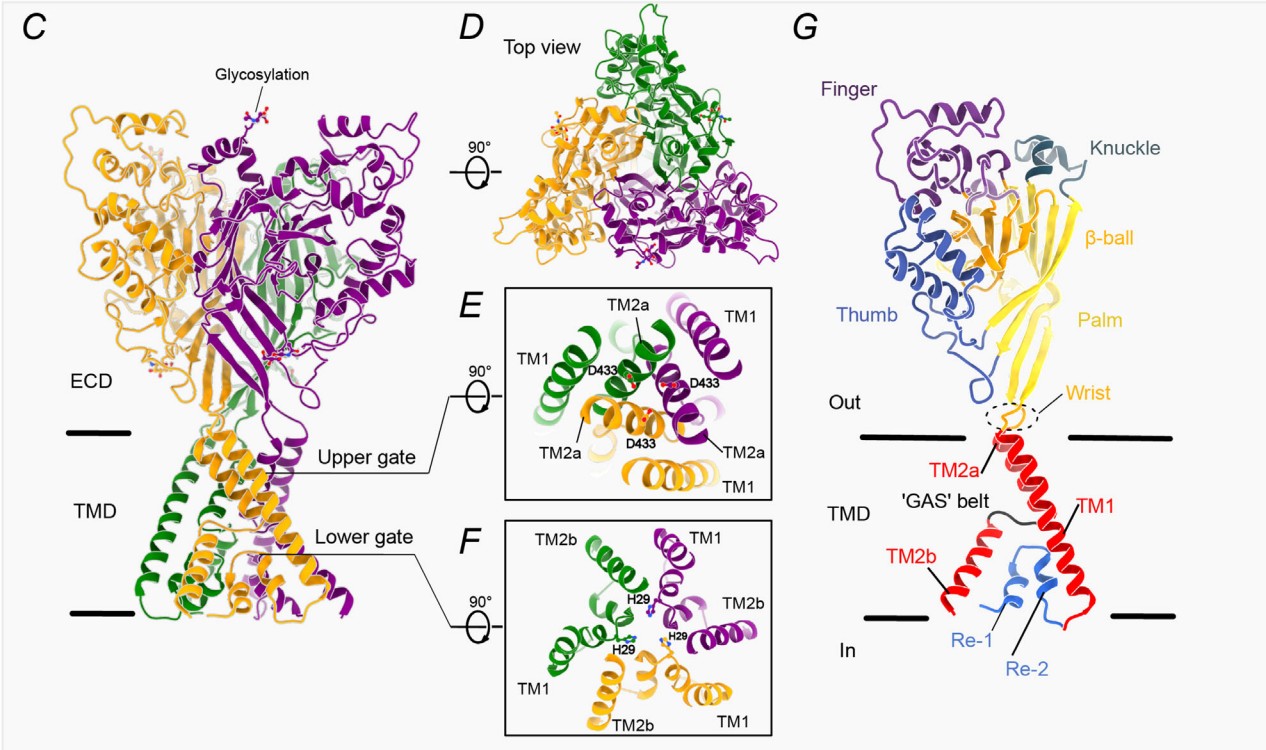

**Figure 1. DEG/ENaC/ASIC channel structure**

*A*, schematic representation of subdomain architecture of an acid-sensing ion channel (adapted from Grunder & Chen, 2010). *B*, schematic representation of trimeric channel structure of chicken ASIC1, where the individual

subdomains are coloured as in *A*. For easier visualization, a subdomain architectural schematic representation is shown for only one subunit. Each subunit has a large globular extracellular domain and a transmembrane domain (TMD) consisting of two transmembrane helices (TM1 and TM2). The N- and the C-terminal regions are relatively short in many family members. Key residues in the acidic pocket and at the entrance to the pore, which are crucial for the binding of divalent cations, are shown in the inset. *C*, structure of the homotrimer determined at low pH, thus capturing the channel in the desensitized conformation (Protein Date Bank (PDB): 6VTK; Yoder & Gouaux, 2020). Protein backbone is shown in cartoon representation while glycans are represented as sticks. Individual protomers are coloured orange, green and purple. *D*, top view of the channel, viewed from the extracellular region. *E* and *F*, slice through the TMD displaying the upper gate (*E*) contributed by Asp433 from all three subunits, whereas the lower gate (*F*) is constituted by His29 from each monomer. *G*, cartoon representation of the single monomer from the trimeric channel shown in *C*, where the individual subdomains are coloured as in the schematic representations shown in *A* and *B*. Images representing protein structures were generated with UCSF ChimeraX (Pettersen et al. 2021).

mechanosensation, including DEG-1, which is required for mechanoreceptor currents in nose nociceptor neurons (Geffeney et al. 2011) and forms a homology group with MEC-4, MEC-10, DEL-1 and UNC-8 as well as vertebrate ENaCs (Kaulich et al. 2022*b*), suggesting that mechanosensory capability may be conserved within this cluster.

*Drosophila* also exhibits a wealth of readily assayable mechanosensory behaviours. The multidendritic neurons, like mammalian nociceptors and *C. elegans* PVD and FLP, have elaborate dendritic branching morphology and are required for multiple nociceptive responses and proprioception. Here too, DEG/ENaC family members play multiple roles. In response to a noxious mechanical stimulus, larvae perform a stereotyped rolling escape response, as for noxious thermal and chemical stimuli (Tracey et al. 2003), that requires the class IV multidendritic neurons (mdIVs). Pickpockets Ppk1, Ppk26 and Ppk30 are specifically required for this behavioural response to mechanical noxious stimuli (Gorczyca et al. 2014; Jang et al. 2019; Zhong et al. 2010). They were identified via a combination of reverse genetics (based on homology to known DEG/ENaCs), forward genetic screens for defects in the behavioural response, and Gal4 driver screens for channels expressed in the mdIVs (Adams et al. 1998; Gorczyca et al. 2014; Guo et al. 2014; Hwang et al. 2007; Jang et al. 2019; Mauthner et al. 2014; Zhong et al. 2010). As with the *C. elegans* multidendritic nociceptors, the mdIVs also function in proprioception. However, whereas in PVD, parallel processing of two distinct mechanosensory stimuli could be explained by different subunit composition (Tao et al. 2019), here the same three subunits are implicated (Ainsley et al. 2003; Gorczyca et al. 2014). The explanation could be that channels at different locations within these branched neurons experience a given stimulus differently (Hall & Treinin, 2011) or have different accessory proteins, or simply that different stimuli could elicit different opening states. As with the degenerins, demonstrating a direct role in mechanotransduction, via heterologous expression of Ppks has proven difficult (Gorczyca et al. 2014; Guo et al. 2014; Jang et al. 2019).

## The tethering model: cytoskeleton and extracellular matrix interactions

One reason for the difficulty in demonstrating mechano-sensitivity for heterologously expressed mechanosensors is likely the importance of cellular context and/or morphology in providing the necessary tension on which a mechanical stimulus can act to bring about conformational change. Here, too, *C. elegans* genetics has provided insight. The 'Mec' screen identified other potential components of the mechanosensory apparatus, including microtubule subunits (Fukushige et al. 1999; Savage et al. 1989, 1994) and extracellular matrix components (Du et al. 1996) (Fig. 2*D*), required for MEC-4 localization (Emtage et al. 2004) and mechano-receptor currents (Bounoutas et al. 2009; O'Hagan et al. 2005). UNC-105 function in muscle is likewise dependent on a collagen, LET-2 (Liu et al. 1996), suggesting an analogous organization. An attractive model, then, is that the channel is tethered by the extracellular matrix (ECM) and microtubules (Garcia-Anoveros & Corey, 1996), acting like a gating spring. However, the microtubules do not localize with MEC-4 (Cueva et al. 2007), although MEC-5 does, so while this could hold true for the ECM component, perhaps an alternative intracellular component performs this function. Oocyte-expressed MEC-4-containing channels are relatively immobile (unlike MEC-10 homotrimers) (Chen et al. 2015), suggesting that MEC-4 is indeed anchored by such an interaction.

## ENaC and shear force: salt reabsorption and vascular tone

Mammalian ENaCs are widely expressed (Canessa et al. 1993, 1994; O'Brodovich et al. 1993; Palmer & Frindt, 1986), and this is reflected in the multi-system defects seen in hereditary diseases (Liddle syndrome, type I pseudohypoaldosteronism and cystic fibrosis-like disease) (Hanukoglu, 2017; Hanukoglu & Hanukoglu, 2016; Kashlan & Kleyman, 2011). ENaC plays an essential

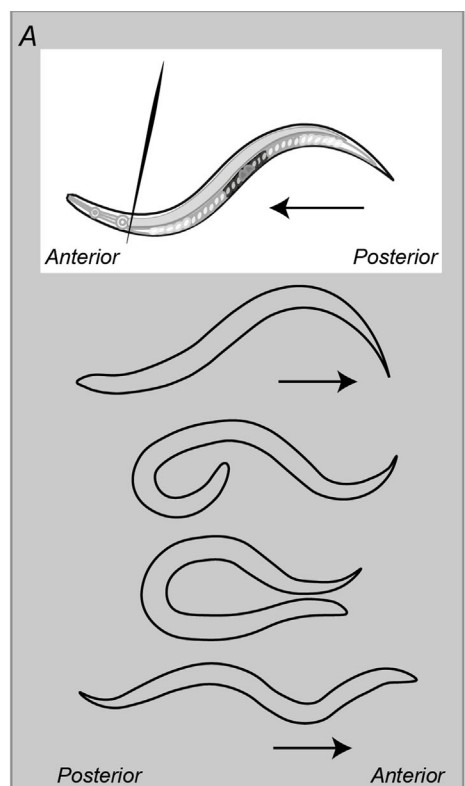

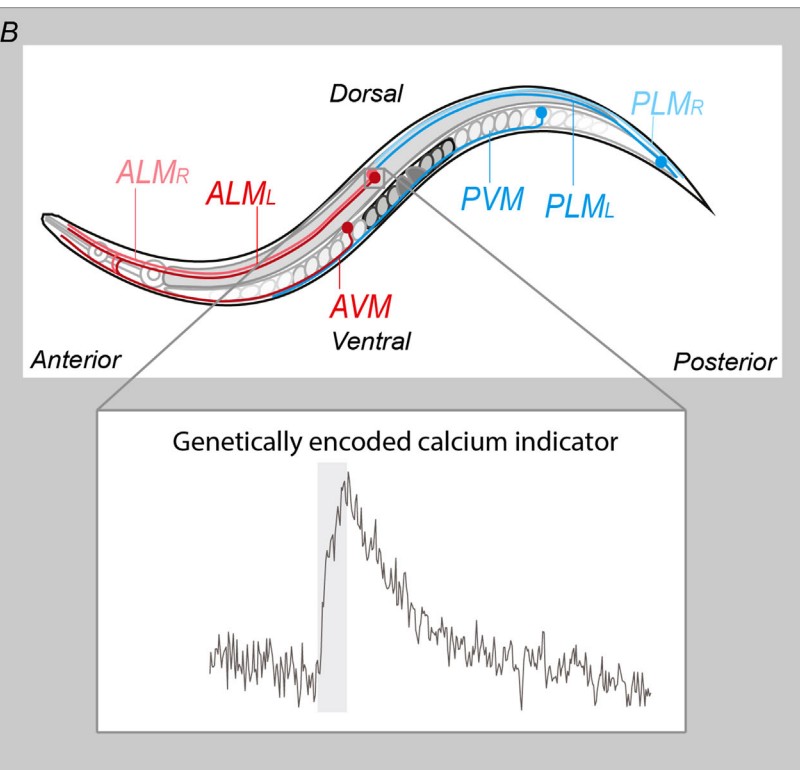

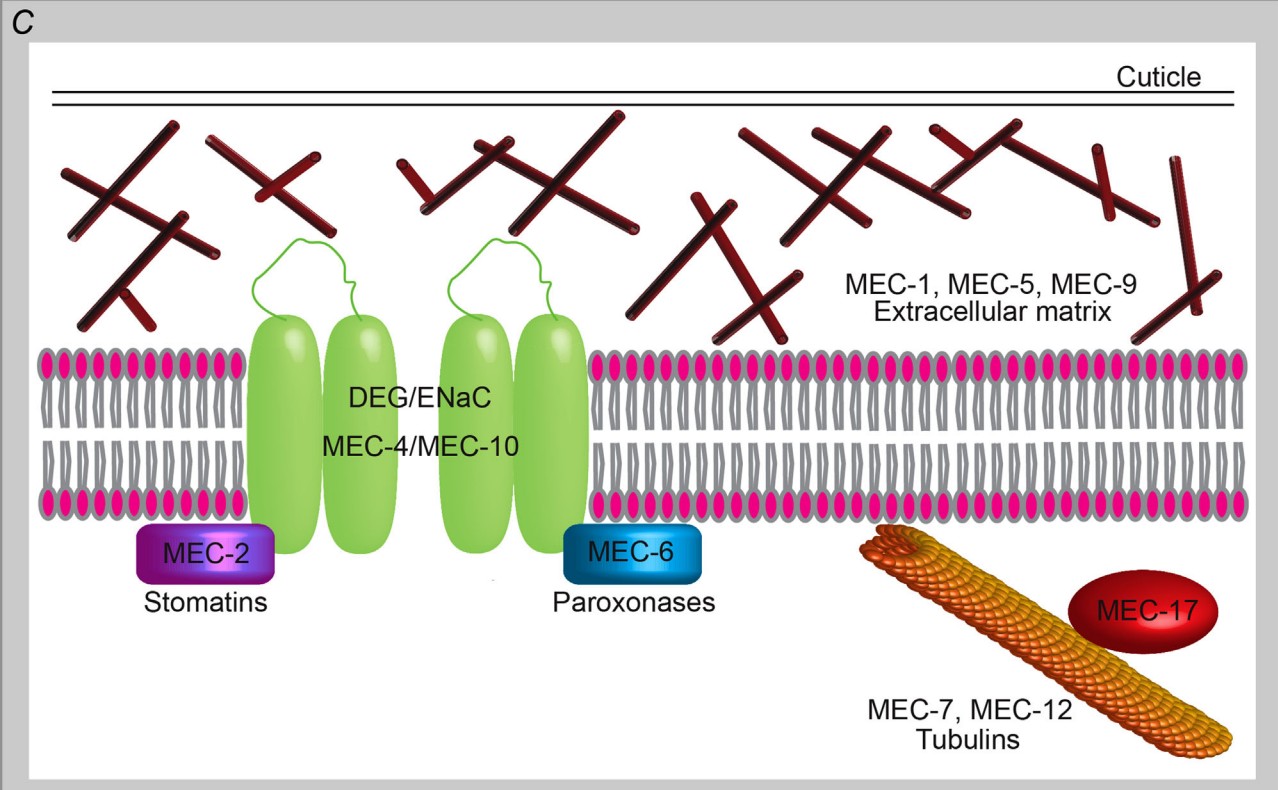

**Figure 2. *C. elegans* gentle touch**

*A*, a behavioural assay for the escape response. When an animal is touched on the anterior body, for example by a hair, it responds by reversing, followed by the execution of a deep turn, enabling it to change the direction of locomotion. Touch on the posterior body produces an opposite response, i.e. accelerated forward movement.

*B*, the touch receptor neurons (TRNs). The TRNs, comprising ALM (anterior lateral microtubule) left and right, AVM (anterior ventral microtuble) and their posterior counterparts, PLM left and right and PVM, are required for the response to gentle body touch. Inset shows a typical calcium transient in ALM, in response to a 1 s gentle touch stimulus (grey bar), recorded in an intact animal using a genetically encoded calcium indicator (in this case, the ratiometric indicator Cameleon; Walker & Schafer, 2020). *C*, presumed components of the mechano-transduction machinery, identified from mapping Mec (gentle touch defective) mutants. *mec-4* and *mec-10* encode DEG/ENaC subunits; *mec-1*, *mec-5* and *mec-9* encode components of the extracellular matrix required for proper MEC-4 localization; *mec-7* and *mec-12* encode microtubule subunits and *mec-2* and *mec-6* encode stomatin and paroxonase family members, respectively, which increase channel currents and appear to function in membrane translocation.

role in controlling $Na^+$ reabsorption in the kidney and the vascular tone of arteries, responding to laminar shear force (Carattino et al. 2004; Karpushev et al. 2010; Satlin et al. 2001), a property shared by ENaCs with subunit compositions different from the canonical $\alpha\beta\gamma$ (Baldin et al. 2020).

The observation that disruption of the ECM disrupts the response to shear force (Knoepp et al. 2020) perhaps points to a MEC-4-like tether model. ENaC interacts with both actin and actin-binding proteins (Karpushev et al. 2010; Mazzochi et al. 2006; Rotin et al. 1994) and disrupting actin decreases ENaC activity (Reifenberger et al. 2014) but does not prevent flow-dependent activation (Karpushev et al. 2010). Actin clearly plays important roles at multiple levels in trafficking of ENaC, regulating its localization to and maintenance at the apical membrane, and thereby regulating $Na^+$ absorption (Butterworth, 2010; Sasaki et al. 2014). There is also evidence for a role for microtubules in ENaC function (Karpushev et al. 2010), but again, flow-dependent activation does not depend on microtubules (Morimoto et al. 2006).

N-linked glycosylation of ENaC plays an important role in surface expression, as well as protease activation (Kashlan et al. 2018). Two glycosylated asparagines are also important for force sensing, and the idea that they function in tethering to the ECM (Knoepp et al. 2020) suggests an interesting avenue of investigation for other mechanosensory DEG/ENaCs. ENaC and MEC-4 also share similarities in regulation by removal from the cell surface, following ubiquitination by the E3 ubiquitin ligases (Butterworth, 2010; Chen & Chalfie, 2015; Ware et al. 2020).

## ASICs function in mechanosensation and proprioception

ASIC knockout phenotypes suggest diverse roles in mechanosensation, including hearing (Wu et al. 2009), blood volume control (Lee et al. 2011), gastrointestinal pain (Jones et al. 2005), and proprioception (Lin et al. 2016). A direct role in mechanosensation has been long debated. A survey of DRG sensory neurons in ASIC2 and ASIC3 knockouts found no evidence of a direct role (Drew et al. 2004). However, this may reflect the population of cells sampled, or the type of stimulus used and, again, the likely need for ECM/cytoskeletal tethering may make demonstrating mechanosensory function challenging. The use of an elastomeric support substrate (and specifically targeting a subpopulation of cells exhibiting prevalence of ASIC3 expression) allowed demonstration of a role for ASIC3 in stretch response in DRG proprioceptive neurons (Cheng et al. 2010; Lin et al. 2016). As we will see below, ASICs have a wider nociceptive role; as key sensors for extracellular protons they play a central role in sensing tissue acidosis and thus in both transmission and sensation of pain. Of particular relevance to chronic pain, and also a confounding factor in distinguishing their precise mechanosensory functions, is their role in inflammation-induced hyperalgesia. For example, in the colon, ASIC3 plays a dual role, in normal visceral mechanosensation of stretch and in its potentiation by inflammation (Jones et al. 2005), whereas in a muscle inflammation model it functions in the muscle to induce cutaneous mechanical hyperalgesia (Sluka et al. 2007).

## Additional accessory proteins modulating DEG/ENaC function

The *C. elegans* 'Mec' screen also identified stomatins and paroxonases, whose conservation across animal models has informed experimental approaches. *mec-2* encodes a homologue of stomatin (Huang et al. 1995); they are founding members of a family of membrane-associated proteins with nine members in *C. elegans* and five in mammals (including stomatin-like STOML1 to 3 and podocin). *mec-2* is required for mechanosensation in the TRNs (Huang et al. 1995; Suzuki et al. 2003) and, like podocin and stomatin, interacts with cholesterol (Huber et al. 2006; Rungaldier et al. 2017). It enhances MEC-4- and MEC-10-derived currents, and mutations that disrupt palmitoylation or cholesterol binding impair its function (Brown et al. 2008; Goodman et al. 2002). Proper localization of the DEG/ENaC channel is also cholesterol-dependent (Cueva et al. 2007; Zhang et al. 2004), indicating a role for MEC-2 in recruiting or maintaining cholesterol in the channel complex. Mammalian ASIC2 and ASIC3 are also modulated by stomatin and STOML3, via direct interactions

(Brand et al. 2012; Lapatsina et al. 2012; Price et al. 2004; Wetzel et al. 2007), suggesting functional conservation. MEC-6, a homologue of paroxonases, also increases MEC-4/MEC-10 currents (Chelur et al. 2002; O'Hagan et al. 2005). Along with a homologue, POML-1, it largely localizes to the endoplasmic reticulum, playing a chaperone-like role, ensuring correct folding and surface expression of MEC-4 (Chen et al. 2016). Mouse paroxonases 2 and 3 play a similar role controlling surface expression of ENaC (Shi et al. 2017; Shi et al. 2020). Again, this suggests functional conservation and illustrates the utility of crosstalk between researchers studying different family members, using diverse experimental approaches.

### A model for understanding degeneration

*C. elegans* degeneration-causing mutations have provided insight into DEG/ENaC channel physiology (Garcia-Anoveros et al. 1995), since they exhibit significant currents when heterologously expressed whereas the wild-type channel does not (Goodman et al. 2002). Degeneration-causing dominant alleles cause constitutive opening of the channel (Driscoll & Chalfie, 1991), leading to unregulated $Na^+$ and $Ca^{2+}$ entry and reactive oxygen species imbalance (Bianchi et al. 2004; Calixto et al. 2012; Hall et al. 1997; Xu et al. 2001). In this way they may exhibit commonalities with hyperactivated $Na^+$, $Ca^{2+}$ permeable ASIC1a channels (Yermolaieva et al. 2004), whose overstimulation by brain acidosis has been proposed to significantly contribute to neuron loss in ischaemia, and which are upregulated at the injury site (Mazzone et al. 2017; William et al. 2020; Xiong & Xu, 2012). However, knock-out or pharmacological blocking of ASICs after *in vitro* spinal cord injury can also intensify excitotoxicity and neuronal death (Mazzone et al. 2017). Consequently, the mechanism(s) by which ASICs are involved is not clear. In contrast to the constitutively active degenerin-causing mutations, ASICs desensitize, limiting the ion flow at prolonged activation, but it has been proposed that acidotoxicity is caused indirectly by, for instance, ASIC C-terminal interactions with the receptor-interacting serine/threonine-protein kinase 1 (RIPK1), triggering phosphorylation of RIPK1, and resulting in neuronal death. *In vivo*, ASICs also localize in close proximity to voltage-gated $Na^+$ and $K^+$ channels, which facilitate transduction of local pH changes into electrical signals and could also be an indirect way to contribute to or protect against neurodegeneration (Mazzone et al. 2017; Petroff et al. 2008; Vukicevic & Kellenberger, 2004).

The readily identified phenotype of the *C. elegans* degeneration mutants has allowed genetic screens for suppressors to be carried out, leading to the identification of genes important for DEG/ENaC function, including *mec-6*, whose homologue paraoxonase is a cellular antioxidant, protecting from oxidative stress (Huang & Chalfie, 1994; Shreffler et al. 1995). These screens have also proven effective for identifying molecular components required for neuronal degeneration that are shared with mammals. For example, Xu and co-workers expressed a *mec-4*(d) allele in motor neurons to induce paralysis and screened for mutations that suppressed degeneration; this screen identified the gene for calreticulin and thus implicated intracellular $Ca^{2+}$ in the degeneration triggered by constitutive MEC-4 activation (Xu et al. 2001). Others have used candidate-based reverse genetics, for example, to elucidate the role of endocytosis and trafficking in neurodegeneration (Troulinaki & Tavernarakis, 2012).

### Acid-sensing members found across phyla

Acidic pH occurs in healthy physiology, such as during synaptic transmission (Du et al. 2014), but is also characteristic of pathological conditions, including tissue injury, infection and inflammation (Riemann et al. 2015). ASICs are important sensors of acidic pH (Ortega-Ramirez et al. 2017; Vina et al. 2013), and there has been extensive interest in their role in healthy and diseased tissue (for detailed reviews see Cheng et al. 2018; Gu & Lee, 2010; Ortega-Ramirez et al. 2017). Insights into acid-sensitive DEG/ENaCs are founded on studies of heterologously expressed channels, with the first ASICs being cloned by the Welsh, Lazdunski and Corey groups (Garcia-Anoveros et al. 1997; Price et al. 1996; Waldmann et al. 1996), followed by the first evidence of activation by low pH (Lingueglia et al. 1997; Waldmann et al. 1997a, 1997b). The structural and functional mechanism of acid sensitivity has been extensively reviewed elsewhere (e.g. Grunder & Pusch, 2015; Ortega-Ramirez et al. 2017; Vullo & Kellenberger, 2020).

Phylogenetic evidence suggests that acid-sensitive DEG/ENaCs evolved in an early bilaterian, serving as an ancient form of receptors for signal transmission (Martí-Solans et al. 2022). Acid-sensitive DEG/ENaCs classify into three main groups. The acid-activated group can be further divided into two subgroups, the first of which is activated by acidic pH (i.e. is completely closed at neutral pH). This includes vertebrate ASICs (Chen et al. 2007; Coric et al. 2005; Springauf & Grunder, 2010; Waldmann et al. 1997b; Zhang & Canessa, 2002), likely *Drosophila* Ppk1 (Boiko et al. 2012) (although this has not been confirmed by heterologous expression), *C. elegans* ASIC-1 and DEL-9 (Kaulich et al. 2022b) and cnidarian, brachiopod, phoronid, annelid, xenacoelomorpha and hemichordate representatives (Aguilar-Camacho et al. 2022; Martí-Solans et al. 2022). The other subgroup of the acid-activated channels includes the human ENaCs

(Collier & Snyder, 2009; Ji & Benos, 2004) and *C. elegans* ACD-2 (Kaulich et al. 2022b), which show some basal activity that can be blocked by amiloride and enhanced by acidic extracellular pH. However, this may not necessarily infer basal activity *in vivo*, depending on the physiological pH environment that the channel encounters, and could also be because heterologous expression rarely reflects the physiological environment. Both subgroups generate large proton-activated inward currents, with channel-specific half-activation pH varying from around 6.5 to 4.3 (Chen et al. 2007; Kaulich et al. 2022b; Waldmann et al. 1997b; Zhang & Canessa, 2002).

The second group, the acid-inhibited channels, display currents at neutral pH and are blocked by decreased pH. These include mouse bile acid-sensitive ion channel (BASIC, also known as ASIC5 or BLINaC) (Wiemuth & Grunder, 2010), *Trichoplax* TadNaC6 (Elkhatib et al. 2019), *C. elegans* ACD-1, ACD-5, DEL-5, DELM-1, UNC-105, heteromeric FLR-1/ACD-3/DEL-5 (Kaulich et al. 2022a, 2022b; Wang et al. 2008, 2012) (DELM, DEgenerin Linked to Mechanosensation; FLR, FLuoRide resistant phenotype), and mutant derivatives of UNC-8 (Wang et al. 2013b). Finally, the third group, mouse BASIC and rat ASIC1a, are actually permeable to protons (Chen & Grunder, 2007; Wiemuth & Grunder, 2010; Wiemuth et al. 2014), the cellular impact of which remains to be elucidated.

Together these channels thus cover a broad pH range, with relevance to a diverse array of biological processes and contexts. For instance, human $\alpha\beta\gamma$ ENaC channels' pH range (minimal at pH 8.5/maximal at pH 6) fits well with kidney collecting duct pH. Similarly, the $pH_{50}$ values of *C. elegans* intestinal channels correspond with their respective apical and basolateral pH environments (Kaulich et al. 2022a), demonstrating the functional relevance of their distinct proton-sensing properties. However, more evidence is needed to interpret their evolutionary relationships and to establish the degree to which acid sensitivity occurs through a common mechanism. The *C. elegans* acid-sensitive members, for example, do not cluster together when overall protein sequence is compared; neither are they the closest homologues of mammalian ASICs (for sequence comparisons see Kaulich et al. 2022b). The extent to which we have so far failed to identify further acid-sensing members, either as homomers or (like FLR-1/ACD-3/DEL-5) as heteromers, is also a pertinent question. Combining this information with more detailed structural and functional analysis, across diverse members, will allow us to better understand the molecular basis of pH dependence. Drawing together channel properties, physiology, cellular function and behaviour is also critical to addressing the precise roles of acid-sensitive members and thus the functional significance of this property.

## Diverse array of modulators of DEG/ENaC channels

With growing interest in the role of DEG/ENaCs in pathology, the list of their *in vivo* and pharmacological modulators has expanded (Kweon & Suh, 2013; Vullo & Kellenberger, 2020; Xiong et al. 2008). For instance, ion concentrations modulate ASIC currents as ASIC gating is tightly regulated by the interplay between protons and cation concentrations (Adams et al. 1999a; Paukert et al. 2004; Sherwood et al. 2011), with significant variability between channel isoforms. In particular, some are permeable to $Ca^{2+}$ (Bässler et al. 2001; Hoagland et al. 2010; Kaulich et al. 2022a; Zhang & Canessa, 2002) or even $Ca^{2+}$-selective, such as the HyNaCs (Dürrnagel et al. 2012). Structural studies, site-directed mutagenesis and electrophysiological characterization have revealed that $Ca^{2+}$ binding at the entrance to the pore can inhibit ASIC1a (Paukert et al. 2004) but can also allosterically modulate other ASICs, changing their response to pH (Babini et al. 2002; Immke & McCleskey, 2003; Zuo et al. 2018). $Zn^{2+}$ is also a potent modulator, blocking or potentiating ASIC2-containing channels and *C. elegans* acid-sensitive members (Adams et al. 1999b; Baron et al. 2001; Jiang et al. 2012; Kaulich et al. 2022b). Structure–function studies could also shed light on whether mechanisms of inhibition might be similar to those of other channel families. For instance, in the $GABA_A$ receptor, a pentameric ligand-gated ion channel, three histidine (His267) residues within the M2 pore-lining helices of the $\beta$ subunits coordinate $Zn^{2+}$ across the pore (Kasaragod et al. 2022), and this structurally resembles the His29 residues of chicken ASIC1a (Yoder & Gouaux, 2020) (Fig. 1).

The neurotransmitter serotonin, an inflammatory mediator, enhances acid-evoked sustained currents of ASIC3-containing channels. This is corroborated by behaviour experiments in which co-application of serotonin enhanced the 'paw-licking' pain response to acid administration, a response that is reduced in *asic3* mutants and inhibited by amiloride (Wang et al. 2013a). The presence of lactate, released along with protons during muscle stress, enhances acid-induced currents of ASIC1a/ASIC3 channels by chelating extracellular free $Ca^{2+}$ and thereby removing the $Ca^{2+}$ block (Immke & McCleskey, 2001, 2003). The ASIC channel thus acts as a 'coincidence detector,' and in cardiac muscle ischaemia the result is increased sensitivity to this local lactic acidosis, and thus chest pain (whereas systemic acidosis resulting in a similar pH does not produce chest pain).

Many pharmacological blockers have also been discovered; the most widely used is amiloride, an anti-hypertensive, and its derivatives, which are potent and specific DEG/ENaC blockers and thus useful pharmacological tools (Bentley, 1968; Canessa et al. 1994;

Palmer & Frindt, 1986). Structural and mutational studies have revealed that it can bind within the channel pore (blocking the channel), as well as in the extracellular acidic pocket (which is thought to result in the paradoxical stimulation of some family members) (Baconguis et al. 2014; Kellenberger et al. 2003; Schild et al. 1997). Amiloride inhibits a wide variety of DEG/ENaCs, with dramatic variation in potency (Boiko et al. 2012; Canessa et al. 1994; Garcia-Anoveros et al. 1998; Goodman et al. 2002; Han et al. 2013; Waldmann et al. 1997b; Wang et al. 2008). Indeed, for some ASICs, it can enhance currents and even activate the channel (Adams et al. 1999b; Kaulich et al. 2022a, 2022b; Li et al. 2011b). ASICs are also targets of non-steroidal anti-inflammatory drugs (NSAIDs) (Voilley, 2004; Voilley et al. 2001), a diverse group of chemicals used to treat pain and inflammation. Their action is highly isoform specific; ASIC1b and ASIC2a are unaffected by most NSAIDs, while ASIC3 is inhibited by salicylic acid and diclofenac and ASIC1a is inhibited by flurbiprofen and ibuprofen (Voilley et al. 2001). NSAIDs also act on *C. elegans* DEG/ENaCs (Fechner et al. 2021), further illustrating the likely structural conservation between nematode and vertebrate channels.

## Neuropeptides and toxins: channel modulation and therapeutic possibilities

In 1990, Cottrell and colleagues identified FMRFamide (Phe-Met-Arg-Phe-amide)-gated ion channels in the snail, *Helix aspersa* (Cottrell et al. 1990). Subsequent investigation identified an entire family of FMRF-amide-gated channels in *Hydra*, the HyNaCs (Assmann et al. 2014; Dürrnagel et al. 2012; Golubovic et al. 2007), as well as additional amidated peptide-gated sodium channels (FaNaCs) from molluscs and annelid channels, gated by FMRFamide, FVRIamides and Wamides (Dandamudi et al. 2022; Lingueglia et al. 1995; Schmidt et al. 2018). Despite extensive screening (Vyvers et al. 2018), no native peptide agonist for mammalian ASICs has yet been identified, but several are known to potentiate acid-induced currents. For example, neuropeptide FF and FMRF-amide slow desensitization of ASIC1a and 1b; endomorphin-1 and -2 have a similar effect on ASIC3 (Askwith et al. 2000; Farrag et al. 2017). This is interesting because although ASICs desensitize rapidly, they are implicated in processes requiring a sustained current, such as inflammatory pain and ischaemic damage. Peptidergic modulation may therefore provide a mechanism for achieving such sustained channel opening. Of note, in the screen of rat neuropeptides, Vyvers et al. (2018) screened 109 of the 294 neuropeptides in the NeuroPep database (Wang et al. 2015), and specifically selected for shorter peptides, so there remains potential for further discoveries. Modulation by neuropeptides may be more relevant in some species than others, and tractable organisms such as *C. elegans*, with abundant neuropeptides, certainly merit investigation. A significant barrier is the cost of such screens, and here genetics may provide alternatives, or a means to pre-screen for likely candidates.

In addition to endogenous neuropeptides, ASICs are targeted by peptide toxins from venomous species, and these are valuable tools for pharmacological and structural investigation, especially because these toxins show high specificity for certain subunits (Cristofori-Armstrong & Rash, 2017). For example, black mamba Mambalgin-1 reversibly inhibits ASIC1a and ASIC1b (Diochot et al. 2012, 2016). Sea anemone APETx2, on the other hand, inhibits ASIC3 (Diochot et al. 2004), although it exhibits promiscuity, in that it also inhibits some voltage-gated sodium channels (Blanchard et al. 2012; Peigneur et al. 2012), limiting its usefulness. Intriguingly, tarantula Psalmotoxin-1 blocks ASIC1a-containing homomeric but not heteromeric channels (Escoubas et al. 2000; Joeres et al. 2016; Sherwood et al. 2011), but at high concentration can promote opening of ASIC1b channels. The explanation lies in its affinity for different channel states, with the toxin stabilizing the desensitized state for ASIC1a and the open state for ASIC1b (Chen et al., 2005, 2006; Escoubas et al. 2000; Sherwood et al. 2011). Finally, peptide toxins can act as agonists. Coral snake MitTx is a potent agonist for ASIC1a (Bohlen et al. 2011), via interaction with the wrist, palm and thumb domains (Baconguis et al. 2014), causing intense, unrelenting pain. For many toxins the exact interaction site has not been established. Further structural studies will be instrumental in providing insights into their mode of action, as well as opening up possibilities for therapeutic use, especially as analgesics (Diochot et al. 2016). More information on structural insights into the mechanisms of action of these and other ligands, with a focus on ASIC1a, can be found in Tikhonov et al. (2019).

From this brief and incomplete survey, it is clear that DEG/ENaCs activity can be influenced by a vast array of modulators, in a highly subunit-specific manner. While *in vitro* expression has provided significant insight into the consequences for channel properties, genetic evidence for the precise role of these modulations in specific cellular contexts *in vivo* remains more limited. The current challenge is therefore to relate the *in vitro* gathered data to in-depth genetic and behavioural studies to fully understand how these channels function *in vivo*.

## Roles of DEG/ENaC channels in taste

A number of DEG/ENaC channels have also been implicated as molecular sensors for taste, but evidence of a direct role is less convincing. ASIC1 and ASIC3 are expressed in mouse and rat taste buds and the lingual epithelium (Richter et al. 2004; Shimada et al. 2006;

Ugawa et al. 2003), and zebrafish zASIC2 and zASIC4 in taste sensory cells (Levanti et al. 2016; Vina et al. 2013). These expression patterns inspired two attractive hypotheses; firstly, since ASICs are acid sensing, that they are sour-taste receptors (Shimada et al. 2006; Ugawa et al. 2003), and secondly, since salt-taste is blocked by amiloride (Spector et al. 1996), that they are salt-taste receptors. Likewise, several invertebrate family members could, on the basis of their expression patterns, be candidates for roles in taste. *C. elegans* DEG-1 is expressed in chemosensory neurons, functioning in responses to both attractive and repellent cues (Wang et al. 2008), and DEL-7 and DEL-3 are enriched in neurons that drive behavioural responses to food localizing close to the site of bacterial lysis, where they could directly sense bacterial components (Rhoades et al. 2019). *Drosophila ppk* genes are expressed in taste-sensing organs and gustatory neurons involved in courtship (L. Liu et al. 2003; T. Liu et al. 2012, 2018; Thistle et al. 2012; Toda et al. 2012; Vijayan et al. 2014) and water sensing (Cameron et al. 2010; Chen et al. 2010).

*C. elegans deg-1* mutants show defective acid avoidance (Wang et al. 2008), supporting a role in acid *sensing*. Evidence for ASIC2 is more disappointing, as knock-out mice exhibit normal responses to acid (Richter et al. 2004). Calcium imaging revealed that *Drosophila ppk28*-expressing cells responded to citric acid taste, but this is osmolarity-dependent; Ppk28 actually forms an osmosensitive channel, essential for water taste (Cameron et al. 2010; Chen et al. 2010).

Several studies also implicate DEG/ENaC channels in salt taste, particularly with regard to low concentrations of sodium. Conditional knock-out of ENaCα subunits in mouse taste receptors selectively abolished NaCl attraction, as well as neuronal responses to low NaCl concentrations (Chandrashekar et al. 2010), but their direct involvement in salt taste remains controversial (for a review, see Bigiani, 2020). RNA interference or expression of a dominant-negative gene for *Drosophila* Ppk11 and Ppk19 disrupted discrimination of low salt concentrations and attenuated neuronal responses (Liu et al. 2003). *Drosophila ppk23* is also expressed in salt gustatory receptor neurons, but a direct taste role was not investigated (Jaeger et al. 2018). Thus, genetic studies have provided evidence that DEG/ENaCs are involved in salt taste and, to a lesser extent, acid taste in a variety of organisms, but how exactly they contribute remains to be seen.

### *Drosophila* Ppk channels and pheromone sensing

Several Ppk channels from *Drosophila* are involved in courtship behaviour. For example, *ppk25* (Lin et al. 2005), *ppk29* and *ppk23* (Thistle et al. 2012; Toda et al. 2012) are enriched in the male foreleg gustatory receptor neurons which recognize female pheromones (Fig. 3). Loss of Ppk23 and Ppk29, or these neurons, promotes courtship of females and inhibits courtship of males (Starostina et al. 2012b; Thistle et al. 2012). The same Ppks function in female receptivity; antenna-less *ppk25*, *ppk23* or *ppk29* mutants are unreceptive to males (Vijayan et al. 2014). Ectopic expression of the three genes conferred the ability to respond to female aphrodisiac pheromone, suggesting that they form a female pheromone receptor (Liu et al. 2018). However, Ppk25 is also expressed in olfactory pheromone receptor neurons (Starostina et al. 2012a), where it appears to function in signal amplification. Pheromone response of these neurons is not completely disrupted in *ppk25* knockout males but fails to increase with age (Ng et al. 2019). Pheromone-sensing properties could be unique to the Ppks, especially given their phylogenetic separation; this would be an interesting avenue to explore in other species.

### Diverse roles for DEG/ENaCs in modulating synaptic function

Across species, DEG/ENaC knock-out mutants show deficits in neuro-modulation and synaptic plasticity, which underlies learning and memory. For example, ASIC1a is expressed in the rat nucleus accumbens, known for its role in addiction linked to associative learning; ASIC1a overexpression has been shown to decrease addictive behaviours, while knock-out increases them (Kreple et al. 2014). ASIC1 localizes postsynaptically, and low pH elicits postsynaptic ASIC currents in murine brain slices. These were abolished in *asic1* knock-outs, which showed defects in associative and spatial learning (Du et al. 2014; Wemmie et al. 2002).

*C. elegans* ASIC-1, which is not, as the name might imply, the direct species orthologue of mammalian ASIC1), has also been shown to be required for associative learning. In contrast to the mammalian ASIC1, worm ASIC-1 localizes presynaptically, and thus it has been hypothesized that protons released along with dopamine from presynaptic terminals might decrease local pH and thus activate ASIC-1, which in turn would promote sustained dopaminergic signalling. Dopamine release was indeed shown to be reduced in *asic-1* mutants, and *asic-1* learning deficits were mimicked by eliminating dopaminergic signalling (Voglis & Tavernarakis, 2008). The recent observation that ASIC-1 is acid-activated (Kaulich et al. 2022b) is in agreement with this hypothesis. Presynaptic roles have been identified for other DEG/ENaC channels. For example, in *C. elegans* nociceptor neurons, dendritic DEL-1, MEC-10 and UNC-8 modulate neuronal activity by inducing local $Ca^{2+}$ increase and neuropeptide release (Tao et al. 2019). Likewise, *in vivo* voltage-clamp recordings from

*Drosophila* mdIVs showed that Ppk1 also responds to acid, conducting a transient depolarizing current sufficient to elicit a burst of action potentials (Boiko et al. 2012). Based on a subsequent observation that Ppk1 localizes presynaptically (see next paragraph; Orr et al. 2017), this points to a potential presynaptic role, modulating neurotransmitter release. Although the functional contexts are different, there may be mechanistic parallels in how these different channels act presynaptically to modulate synaptic function.

There is also evidence for roles for DEG/ENaCs in synaptic homeostasis. An electrophysiology-based genetic screen in *Drosophila* searched for mutants defective in homeostatic increase of presynaptic release induced by blocking postsynaptic glutamate receptors; this screen identified mutations in two DEG/ENaC channel genes,

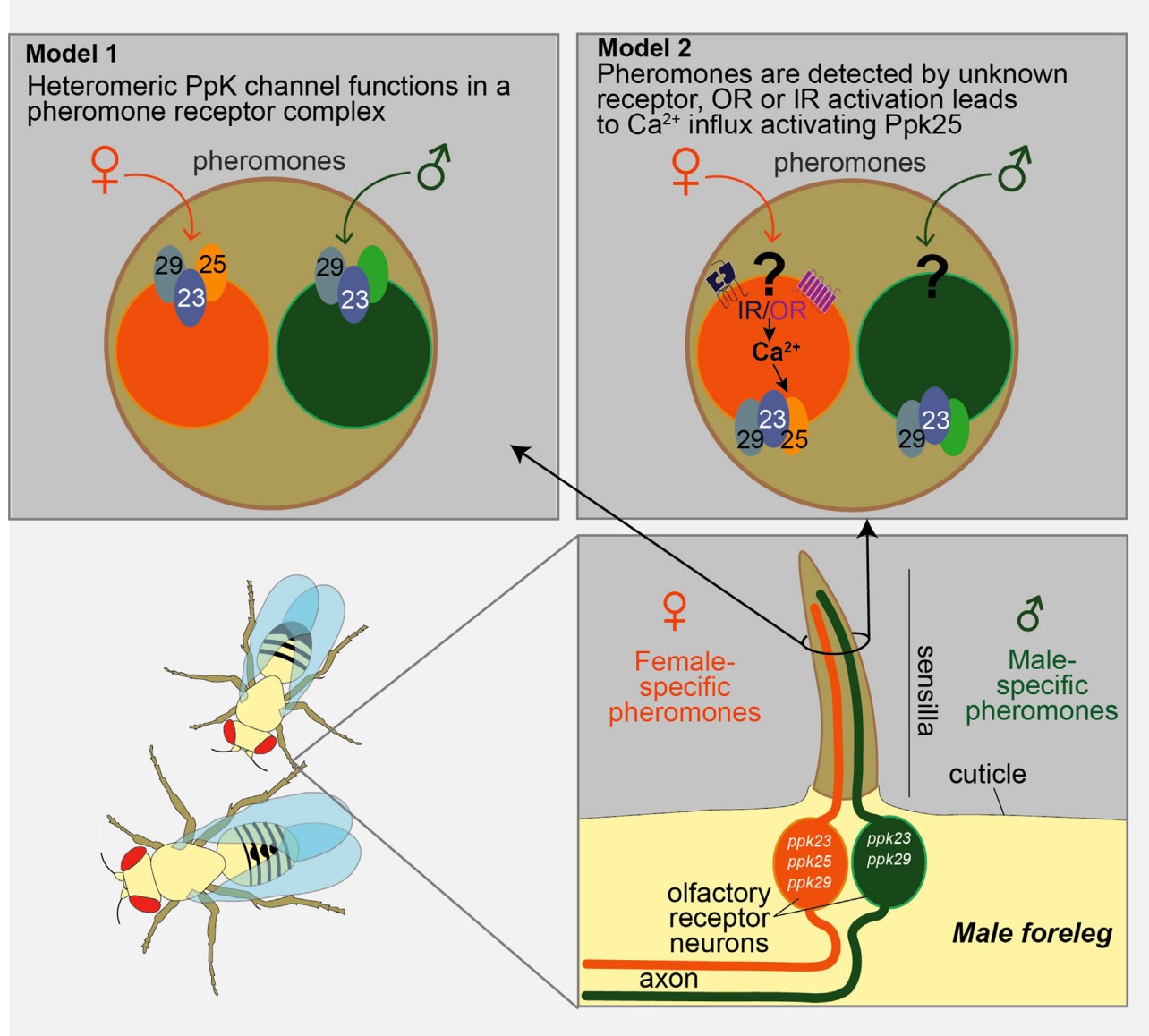

**Figure 3. Ppk modulation of *Drosophila* courtship behaviour**
Contact detection of pheromones by gustatory hairs on the male foreleg. The hairs contain pheromone-sensing gustatory neurons, expressing Ppks, with complementary response profiles: F cells (orange) detect female pheromones, stimulating male–female courtship; M cells (green) detect male pheromones, inhibiting male–male courtship. F cells express Ppk25/Ppk29/Ppk23 heteromeric channel; in M cells Ppk25 is thought to be replaced by an unknown subunit. Two alternative models for the role of Ppk channels are presented. In model 1, the Ppk25/Ppk29/Ppk23 heterotrimer directly senses female pheromone. In model 2, unknown olfactory receptors detect pheromone and their activation leads to $Ca^{2+}$ influx, which in turn activates the Ppk channel complex, leading to signal amplification (based on Pikielny, 2012).

*ppk11* and *ppk16* (Younger et al. 2013). The products of these genes form a $Na^+$ leak channel that generates a subthreshold depolarization, indirectly increasing action potential-induced $Ca^{2+}$ influx and thus neurotransmitter release; a subsequent RNAi screen added Ppk1 as the third subunit of the putative channel (Orr et al. 2017). Similarly, postsynaptic *Ppk29 h*as been shown to regulate the neuromuscular junction via alteration of glutamate receptor properties (Hill et al. 2017). Whether this latter function relies on (co-released) proton sensing is not clear.

Finally, DEG/ENaCs can also exert an effect on neuronal function from surrounding glia. *C. elegans* offers two examples. Mutation of *acd-1*, expressed in the amphid glia, exacerbates sensory deficits caused by mutations of genes implicated in amphid sensory functions, including *deg-1*. Artificially increasing intracellular $Ca^{2+}$ levels in the sensory neurons bypassed the need for ACD-1, suggesting ACD-1 modulates neuronal excitability (Wang et al. 2008, 2012). DELM-1 and DELM-2 exert a similar effect in the glia associated with the OLQ and IL1 nose touch neurons (Han et al. 2013). Expression of vertebrate ASIC genes has also been described in a variety of glia cell types (for reviews see Cegielski et al. 2022; Hill & Ben-Shahar, 2018) and, for example, some of the roles identified in learning may include a glial component or indeed be glia-based, given the impact of glia on synaptic plasticity (Sancho et al. 2021; Singh & Abraham, 2017). Whereas in flies and worms cell-type-specific rescue or knockout is routine, most studies using knockout rodents are confounded by absence of the gene from both neurons and glia. So, disentangling glial from neuronal functions represents an exciting, challenging, avenue for future investigations. In summary, with the help of genetic, behavioural and electrophysiological approaches, we have seen that there is strong evidence for a wide variety of roles for DEG/ENaCs in the modulation of neuronal and synaptic function. We suggest that combining these approaches together with comparative analysis will be essential to understand the diversity of this super-family as well as unique and common features among the members.

## Concluding remarks

We have seen that the DEG/ENaC superfamily is a diverse family of ion channels that function in a variety of tissues and participate directly or indirectly in a vast array of cellular responses and behaviours. Their properties vary greatly, and a huge diversity of subunit-specific modulators further fine-tune these properties, adding a further layer of complexity. We have attempted to identify overarching themes across family members, and indeed see commonalities, such as the conservation of pH

sensitivity, mechanosensitivity and the roles of accessory and cytoskeletal proteins. However, an important caveat is that without more extensive characterization it is unclear to what extent these themes are universal. Many channel types exhibit some degree of pH sensitivity, for example: so are all DEG/ENaC family members in fact pH sensitive, or capable of forming pH-sensitive heteromers, or is this property unique to specific sub-branches? Likewise, for mechanosensitivity; a wide range of channels exhibit mechanosensitivity or an undefined role in mechano-sensation: is this a common characteristic of DEG/ENaCs, for which we simply have not widely tested, or have not found an appropriate stimulation protocol, or can we identify specific residues or structural features that enable this property in a subset of members? Higher throughput, automated electrophysiological techniques may help to address these questions. The second major gap in our understanding is that for any given family member our information is incomplete. We may have a wealth of information gleaned from one particular type of approach, such as analysis of channel properties or genetic analysis of function, but lack the information from other experimental approaches to enable us to relate molecular function with cellular and whole animal function. For different family members, the emphasis has been on different specific approaches, making direct comparison across phyla difficult.

Many DEG/ENaCs have been shown to respond to multiple factors. One hypothesis is that the cellular context determines their functional relevance. For example, *C. elegans* DEG-1 functions as a mechanoreceptor in the ASH nociceptor neurons, in acid avoidance and lysine attraction in the ASK neurons, and as a thermosensor in the ASG neurons (Takagaki et al. 2020). Another hypothesis is that different subunit compositions perform distinct roles within the same neuron. In the *C. elegans* PVD neurons, for example, distinct DEG/ENaC subunit combinations function in proprioception and nociception (Tao et al. 2019). We have also seen that function can change with development, which to our knowledge has only be studied in *Drosophila*. Ppk29 and Ppk11 function at neuromuscular junctions in larval locomotion, whereas in the adult, Ppk29 is implicated in pheromone sensing and *ppk11* in salt sensing. Finally, a recurring theme has been that one modulator potentiates the response to another; for example, while vertebrate ASICs primarily respond to changes in extracellular proton concentrations, many other factors can tune these responses. ASICs are thus acting as coincidence detectors, eliciting a response, or a particular type of response, only in the presence of a combination of factors. This in itself helps to explain how a single subunit type can have apparently diverse functions in different cellular contexts, and suggests that we will see many more examples of DEG/ENaCs performing complex roles in synaptic regulation and plasticity.

As a result of functional conservation, invertebrate models can help to shed light on our understanding of the clinically important ASICSs and DEG/ENaCs, and vice versa. However, the diversity of the family also has a significant role to play in furthering our understanding of channel properties, via comparative structure–function analysis, combined with an understanding of their cellular context. The power of genetically amenable models lies not just in the ability to carry out large-scale screens, but, more specifically, to tackle questions at multiple levels. *C. elegans* and *Drosophila*, in particular, have been at the forefront of genetic and optogenetic innovation, allowing dissection of the role of a gene at the level of an individual neuron. When this is combined with an understanding of channel properties, from *in vitro* studies, we start to unlock the full potential of these models. While a small handful of the invertebrate candidates exemplify this multi-level approach, a large number of family members remain to be fully characterized. They represent significant untapped potential in helping to understand the physiological and cellular basis of diverse DEG/ENaC functions.

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

## Additional information

### Competing interests

The authors have no competing interests to declare.

### Author contributions

All authors approved the final version of the manuscript and agree to be accountable for all aspects of the work in ensuring that questions related to the accuracy or integrity of any part of the work are appropriately investigated and resolved. All persons designated as authors qualify for authorship, and all those who qualify for authorship are listed.

### Funding

This work was supported by the Medical Research Council, as part of United Kingdom Research and Innovation (also known as UK Research and Innovation) [MRC file reference number MC-A023-5PB91], by the Wellcome Trust [Grant reference number WT103784MA] and by the National Institutes of Health [grant reference numbers R01NS110391 and R21DC015652], all to W.R.S. For the purpose of Open Access, the MRC Laboratory of Molecular Biology has applied a CC BY public copyright licence to any Author Accepted Manuscript (AAM) version arising from this submission.

### Acknowledgements

The authors are very grateful to Vikram Kasaragod for helpful discussions, advice on structural representations and edits on the manuscript. Molecular graphics and analyses were performed with UCSF ChimeraX, developed by the Resource for Biocomputing, Visualization, and Informatics at the University of California, San Francisco, with support from National Institutes of Health R01-GM129325 and the Office of Cyber Infrastructure and Computational Biology, National Institute of Allergy and Infectious Diseases.

### Keywords

acid-sensing ion channels, DEG/ENaCs, degenerins, epithelial $Na^+$ channels, pickpockets

### Supporting information

Additional supporting information can be found online in the Supporting Information section at the end of the HTML view of the article. Supporting information files available:

**Peer Review History**

