## [Peer Review History · The Journal of Physiology]

The diverse functions of the DEG/ENaC family: linking genetic and physiological insights

Eva Kaulich, Laura J Grundy, William R Schafer, and Denise S Walker
DOI: 10.1113/JP283335

Corresponding author(s): Denise Walker (dwalker@mrc-lmb.cam.ac.uk)

The following individual(s) involved in review of this submission have agreed to reveal their identity: Stephan Kellenberger (Referee #1)

Review Timeline:

Submission Date:	06-Jul-2022
Editorial Decision:	02-Aug-2022
Revision Received:	22-Sep-2022
Editorial Decision:	07-Oct-2022
Revision Received:	13-Oct-2022
Accepted:	25-Oct-2022

Senior Editor: Laura Bennet

Reviewing Editor: Morag Mansley

Transaction Report:

Dear Dr Walker,

Re: JP-TR-2022-283335 "The diverse functions of the DEG/ENaC family: linking genetic and physiological insights" by Eva Kaulich, Laura J Grundy, William R Schafer, and Denise S Walker

Thank you for submitting your Topical Review to The Journal of Physiology. It has been assessed by a Reviewing Editor and by 2 expert referees and I am pleased to tell you that it is considered to be acceptable for publication following satisfactory revision.

The reports are copied at the end of this email. Please address all of the points and incorporate all requested revisions, or explain in your Response to Referees why a change has not been made.

NEW POLICY: In order to improve the transparency of its peer review process The Journal of Physiology publishes online as supporting information the peer review history of all articles accepted for publication. Readers will have access to decision letters, including all Editors' comments and referee reports, for each version of the manuscript and any author responses to peer review comments. Referees can decide whether or not they wish to be named on the peer review history document.

I hope you will find the comments helpful and have no difficulty in revising your manuscript within 4 weeks.

Your revised manuscript should be submitted online using the links in Author Tasks Link Not Available. This link is to the Corresponding Author's own account, if this will cause any problems when submitting the revised version please contact us.

You should upload:

- A Word file of the complete text (including any Tables);
- An Abstract Figure, (with accompanying Legend in the article file)
- Each figure as a separate, high quality, file;
- A full Response to Referees;
- A copy of the manuscript with the changes highlighted.
- Author profile. A short biography (no more than 100 words for one author or 150 words in total for two authors) and a portrait photograph of the two leading authors on the paper. These should be uploaded, clearly labelled, with the manuscript submission. Any standard image format for the photograph is acceptable, but the resolution should be at least 300 dpi and preferably more.

- A 'Cover Art' file for consideration as the Issue's cover image;
- Appropriate Supporting Information (Video, audio or data set https://jp.msubmit.net/cgi-bin/main.plex?form_type=display_requirements#supp).

To create your 'Response to Referees' copy all the reports, including any comments from the Senior and Reviewing Editors into a Word, or similar, file and respond to each point in colour or CAPITALS. Upload this when you submit your revision.

I look forward to receiving your revised submission.

Yours sincerely,

Professor Laura Bennet
Senior Editor
The Journal of Physiology
<https://jp.msubmit.net>
<http://jp.physoc.org>
The Physiological Society
Hodgkin Huxley House
30 Farringdon Lane
London, EC1R 3AW
UK
<http://www.physoc.org>
<http://journals.physoc.org>

EDITOR COMMENTS

Reviewing Editor:

The review of the manuscript, "The diverse functions of the DEG/ENaC family: linking genetic and physiological insights" that was submitted to The Journal of Physiology is complete, having been assessed by 2 expert referees as well as the reviewing and senior editors.

The Editors have carefully read your manuscript and considered the points raised by the referees. The points raised by both referees should be addressed, in particular correcting statements regarding structure/function and the initial cloning of ASIC (1a not 3), as well as refining the focus of the review and elaborating on current conflicts in published data on ENaC/DEG. A legend for the abstract figure is also required, please see guidelines for the Journal at <https://bit.ly/3sVoley>.

Senior Editor:

Thank you for your review. Both reviewers find merit in the review, but note significant/major amendments required to ensure accuracy, depth of knowledge and a good flow for the reader. Of note, there is a need to consider a tighter focus for the review as currently it is currently very broad coming across as superficial in places and insufficiently balanced in depth, which is a key focus of reviews in The Journal. A stronger focus would increase its potential influence.

REFEREE COMMENTS

Referee #1:

This manuscript gives an interesting overview of the functions and roles of different members of this large ion channel family. It contains a lot of information and is well written. My remarks concern mainly some statements on structure-function aspects that are not entirely correct, and then I would like to have some passages clarified.

1. Summary, line 5, "ENaCs and ASICs...". The roles of ENaC and ASICs are very different. It might be better to separate them in two sentences, because in this sentence it is not clear which of the mentioned roles are attributed to ENaC and which to ASICs.
2. 2nd paragraph on p.4, line 4, "Both form a homotrimer...". The high resolution structures obtained so far are from ASIC1a homotrimers; however ENaC was expressed as heterotrimer of alpha, beta and gamma subunits. This should be corrected. As a reference for the ENaC structure, Noreng et al., 2020 may be more appropriate than the 2018 article, since it is more complete.
3. A little bit further down, "The structure of each protomer..", the central, beta-strand-rich scaffold is formed by the palm, not the beta-ball.
4. Pore, gate, selectivity filter (bottom of p.4 and top of p.5). You should be careful to not mix up "gate" and "selectivity filter". The gate at the level of Asp433 is not put into question by the newer studies. It was never stated, contrary to what you mention, that Asp433 contributed to ion selectivity. The "GAS-belt" is mainly involved in ion selectivity. The structural article by Yoder and a functional study (doi:10.7554/eLife.24630) questioned somewhat this role, identifying acidic residues further down the pore contributing to the selectivity. A recent study that you cite (Chen et al., 2022) suggested then that the effect of these lower acidic residues on ion selectivity is indirect, via the GAS-belt. It would be good to rewrite this part on gating and ion selectivity. Regarding the last sentence in this paragraph, the Na, K or Ca selectivity, I was not aware of K-selective members. Are these really K-selective, or just non-selective cation channels?
5. Fig. 1. This is a nice and clear figure. The calcium ions are shown as "2+". Could you mark them as "Ca2+" or at least indicate in the legend that "2+" means Ca ions? You mention in the legend also that the N- and C-termini are short. In ENaC they are 100 residues long or longer. You should relativize this statement.
6. Fig. 2. For the trace in figure 2B, can you indicate which Ca indicator was used, or show a reference?

7. p9, last paragraph, "Thus, in the TRNs,..." Could you better explain what the basis of the mentioned hypothesis is. The sentence following this one would rather suggest the opposite of the hypothesis.

8. p.10, bottom and p11 top, the tethering model. In a study by M. Goodman of 2002 (doi 10.1038/4151039a) it was shown that co-expression of MEC-4 and MEC-10 with MEC-2 was sufficient to express ion currents in *Xenopus* oocytes. So it seems that tethering on both sides is not necessary. Please discuss this evidence in the text.

9. Actin and ENaC (bottom of p11). The paper by Berdiev is based on ENaC measurements in an artificial system. It seems surprising that actin should affect the single-channel conductance. Could you provide more information with this statement, or take it out?

10. p12, "ASIC function in noxious mechanosensation...". Several aspects discussed in this paragraph are not really noxious functions. Could you adapt the title? You mention a role of ASIC3 in stretch response. The indicated reference is a method paper, and I did not find any mention of ASIC3. Could you double-check it? An important article on the unsuccessful cellular measurement of an involvement of ASICs in mechanosensation is the article by Drew and Wood (PMID: 14990679). From studies with KO mice there is also evidence for a mechanosensory role of ASICs in the gastrointestinal tract. It might be worth to mention it.

11. p13, " A model for understanding degeneration". In the first lines, some mutations of Deg/ENaC channels in *C. elegans* are mentioned that were used for genetic screens. It would be good to mention that these mutations allow uncontrolled Na entry leading to cell death. This information would help for the understanding of what follows. The explanation is given at the bottom of the paragraph. It would be good to move it up. This would make the text more logical.

12. bottom line of p.13, "...with the rat ASIC3 being the first gene cloned...". This is not correct. ASIC1a and ASIC3 were published in the same year, and since the identification of ASIC1a was published before that of ASIC3, it was published in nature. What was new in these two papers was that the authors had found out that ASICs are activated by low pH. In the years before, several ASICs had been cloned by the laboratories of Michel Lazdunski, Michael Welsh and David Corey, and they had called them "brain Na channel" or similar, and had at this time not found any way of activating them, except by the degenerin mutation. For this reason, it is not correct to state that ASIC3 was the first cloned ASIC.

13. p. 14, different groups of acid-regulated channels. For the "acid- activated" group, it would be useful to distinguish channels that are indeed activated by acid (i.e. from completely closed to open), as the ASICs, and channels whose activity is modulated by pH. To this second group would for example belong the human ENaC, whose basal activity can be changed by +/- 30% depending on the pH.

The BLINAC is in this context named ASIC5. Since this channel is not closer to ASIC than it is to other ENaC/Deg members, why not calling it "BASIC", as recently proposed?

14. p.4, bottom, "In contrast to its derivative benzamil...". amiloride and benzamil are closely related and bind to the same site in ENaC/Deg channels. The observed difference in voltage dependence between the two compounds may be due to different conditions, different channels, etc. If you want to mention this difference, you should probably better describe the conditions of studies investigating amiloride and benzamil.

15. bottom of p.15, "Likewise, lactate...", the reference is not correct, it should be PMID: 11528414 DOI: 10.1038/nn0901-869.

16. p16, lower paragraph, discussion of toxins. Not all the statements in this paragraph are completely correct. Mambalgin inhibits ASIC1a but also ASIC1b. Psalmotoxin inhibits according to most of the studies only ASIC1a homotrimers, but not ASIC1a-containing heterotrimers. It is correct that APETx2 inhibits only ASIC3 of the ASICs, however, it is also an inhibitor

of some voltage-gated Na channels (Peigneur et al., PMID: 22972919 and Blanchard et al., PMID: 21943094). Therefore, the usefulness of APETx2 in animal studies is somewhat limited. I suggest to re-formulate some of the statements.

17. p20, second paragraph, "Likewise, in vivo...". There is not sufficient information shown to really understand the statement. It would be good to add somewhat more information, e.g. indicating in which type of neuron this activity was measured, what was the basis for the conclusion that the role is presynaptic.

18. p21, top paragraph, "Vertebrate ASICs are also expressed in glia...". This sentence gives a wrong impression of the current situation. In fact, there are many papers showing high ASIC currents in neurons and only two papers showing ASIC activity in glial cells of vertebrates, and only one of these papers provided current traces, which showed that the currents in the glial cells are quite small. Many articles addressing learning have clearly shown that the role of ASICs was in the neurons. Therefore, for learning in vertebrates, the current task is not to disentangle neuronal from glial functions, but rather to find whether glial ASICs may also have roles in learning.

Referee #2:

This is an interesting review that summarizes diverse findings on DEG/ENaC ion channels. While the review will be interesting to many readers interested in ion channels, it would still profit from a stronger focus and from a more critical summary of sometimes conflicting published data.

General comments:

1) This review does not have a clear focus. It tries to cover many diverse and unrelated aspects of DEG/ENaC channels. I think it would profit from a clearer focus on a subset of specific aspects, as for example mechanosensitivity or proton sensitivity of DEG/ENaCs.

2) I had the impression that the authors strive to find overarching and common functions for DEG/ENaCs from different species. But it appears that DEG/ENaCs have surprisingly diverse and often still enigmatic functions. Therefore, highlighting common functions and properties across many different species ignores conflicting data and over-simplifies the state-of-knowledge. Moreover, controversial opinions in the field are ignored. But I think the reader would profit from a more balanced and more critical review of different opinions and different findings. The authors paint a picture that is, in my opinion, too black-and-white.

3) Often many findings on DEG/ENaC functions are simply summarized without providing context. Sometimes this suggests conclusions that are misleading. I provide some examples below in my specific comments.

4) "Cross-phylum evidence for ASICs as proton-sensors." In this section, the authors give the over-simplified impression that ASICs in many different phyla act as proton-sensors. This is somewhat simplistic and misleading. The authors should consider that it will be hard to find any ion channel that is NOT in some way influenced by protons. Inhibition by acid, for example, may have many reasons, for example a simple block of the ion pore by permeating protons. This does not prove an evolutionary conserved mechanism.

5) I found the section entitled "Many ligands differentially modulate DEG/ENaC channel activity" particularly weak. The author define a ligand in a very broad sense. From pharmacological inhibitors to cations, serotonin and lactate. The mode of action of all these "ligands" is very different. Lactate actually does not bind to ASICs but simply chelates Ca²⁺. After the next section on "Neuropeptides and toxins", the authors conclude "that DEG/ENaC are modulated by a diverse array of ligands". Are amiloride and NSAIDs really ligands modulating DEG/ENaCs? And why should these drugs or toxins have a

"role in vivo"? This part is more confusing than enlightening.

6) Personally, I found also the concluding remarks too enthusiastic and uncritical. How likely is it really that DEG-1 and PPK29 and PPK11 have so divergent functions? Isn't it also possible (and perhaps more likely) that we still have not understood all their roles well enough?

Specific comments:

1) For the lay reader, please give a brief explanation of the terms "forward genetic screen" and "reverse genetic screen".

2) Page 2: "Protons bind ASICs in this acidic pocket". The essential proton binding sites of ASICs are still unknown. Probably there are many of them and some outside the acidic pocket. This complex view is not really reflected here.

3) I think for some of the figures, it would be fair to mention the sources that they are based on.

4) Page 9: "Pickpockets ... are specifically required for response to mechanical stimuli." What does this mean exactly? Are they mechanosensitive ion channels? Is the evidence unequivocal? What is their specific function for responses to mechanical stimuli?

5) Page 10: "Their [ENaCs] wide-ranging functions are reflected in the multi-system defects seen in hereditary diseases (Liddle syndrome, type I PHA and CF-like diseases)". The defects in these diseases are all related to Na⁺ reabsorption and do NOT reflect wide-ranging functions.

6) Page 10: What are "non-canonical ENaCs"?

7) Page 11: Why does cholesterol-dependence of proper localization of DEG/ENaC channels indicate a role in determining the lipid environment of the channel? Isn't it the opposite?

8) Page 11: What is the relevance of modulation of ASIC2 and ASIC3 by stomatin and STOML3?

9) Page 12: whether the unregulated Na⁺ and Ca²⁺ entry of ASIC1a leads to neuronal death in ischemia is still controversial. In contrast to Mec/DEG genes with constitutively activating mutations, ASICs desensitize, limiting the ion flow at prolonged activation.

10) ASIC3 was NOT the first ASIC gene to be cloned.

11) Page 16: The hypotheses on ASICs as sensors of sour and salt taste are quite old and have not been confirmed.

12) To give a positive example: I found the discussion of the role of PPKs in courtship behaviour interesting and helpful. For example, figure 3 nicely illustrates two alternative models. I would have liked to see more of such balanced view that may

informed future studies.

13) Page 19: *C. elegans* ASIC-1 is unrelated to (not the species ortholog of) mammalian ASIC1. This information is lacking here and the discussion of ASIC-1 is therefore somewhat misleading.

REQUIRED ITEMS:

-Please include a full title page as part of your article (Word) file (containing title, authors, affiliations, corresponding author name and contact details, keywords, and running title).

-Please include a legend to accompany your abstract figure.

-Please upload separate high quality figure files via the submission form.

-It is the authors' responsibility to obtain any necessary permissions to reproduce previously published material
https://jp.msubmit.net/cgi-bin/main.plex?form_type=display_requirements#use

END OF COMMENTS

Confidential Review

06-Jul-2022

22nd September 2022

Dear reviewers and editors,

RE: **JP-TR-2022-283335** The diverse functions of the DEG/ENaC family: linking genetic and physiological insights.

Thank you very much for your constructive critique of our manuscript and insightful suggestions for improvements. We hope that you will agree that our manuscript is much improved, thanks to your input. Please find our response below, where we have addressed these comments individually, outlining in purple the changes that we have made (with line numbers according to the version with tracked changes).

In response to the editors' and referee #2's more general comment regarding the focus, the aim of this review, and indeed our motivation for writing it, was to try to bridge the divide between mammals and invertebrates and between channel physiology and genetics. We felt that this broader view was often missing from the field and could be an obstacle for future advances. We hope that the edits that we have made have served to better explain this objective and to emphasise the focus on bringing together evidence from different animals and different experimental approaches, to highlight common themes and potential mechanistic parallels.

Please note that Kaulich et al. 2022b has been submitted for inclusion in the same special issue. Pending its acceptance, we would then need to replace the current BioRxiv reference with this reference.

Regarding permissions to reproduce figures, we have two figures that are adapted from (but not directly using content from) previously published figures. For figure 1 panel A (adapted from Grunder S & Chen X. (2010). Structure, function, and pharmacology of acid-sensing ion channels (ASICs): focus on ASIC1a. *Int J Physiol Pathophysiol Pharmacol* **2**, 73-94), the journal replied by email that (for non-commercial use) we can reuse the figure for free. For figure 3 (adapted from Pikielny CW. (2012). Sexy DEG/ENaC channels involved in gustatory detection of fruit fly pheromones. *Sci Signal* **5**, pe48), we have uploaded the permission received from the journal.

Best wishes,

Denise Walker.

EDITOR COMMENTS

Reviewing Editor:

The review of the manuscript, "The diverse functions of the DEG/ENaC family: linking genetic and physiological insights" that was submitted to The Journal of Physiology is complete, having been assessed by 2 expert referees as well as the reviewing and senior editors.

The Editors have carefully read your manuscript and considered the points raised by the referees. The points raised by both referees should be addressed, in particular correcting statements regarding structure/function and the initial cloning of ASIC (1a not 3), as well as refining the focus of the review and elaborating on current conflicts in published data on ENaC/DEG. A legend for the abstract figure is also required, please see guidelines for the Journal at <https://bit.ly/3sVoley>.

We have edited the structural section and that describing the initial cloning of ASICs, as outlined in our responses to referee #1's comments 2-4 and 12. We have also edited to define the focus more clearly and to provide a more critical/balanced view, as outlined in our responses to Referee #2's comments 1-2, as well as our general comment in the cover letter above. We have added a legend for the abstract figure (please see figure legends, line 614).

Senior Editor:

Thank you for your review. Both reviewers find merit in the review, but note significant/major amendments required to ensure accuracy, depth of knowledge and a good flow for the reader. Of note, there is a need to consider a tighter focus for the review as currently it is currently very broad coming across as superficial in places and insufficiently balanced in depth, which is a key focus of reviews in The Journal. A stronger focus would increase its potential influence.

We hope that in addressing the individual comments we have improved accuracy, depth of knowledge and flow. Regarding focus and balance, please see our individual replies to Referee 2's comments 1 and 2, and our general comment in the cover letter above.

REFeree COMMENTS

Referee #1:

This manuscript gives an interesting overview of the functions and roles of different members of this large ion channel family. It contains a lot of information and is well written. My remarks concern mainly some statements on structure-function aspects that are not entirely correct, and then I would like to have some passages clarified.

1. Summary, line 5, "ENaCs and ASICs...". The roles of ENaC and ASICs are very different. It might be better to separate them in two sentences, because in this sentence it is not clear which of the mentioned roles are attributed to ENaC and which to ASICs.

We completely agree, and have amended the text to make this clearer. Line 17 now reads: "ENaCs and ASICs have wide ranging medical importance, for example, ENaCs play an important role in respiratory and renal function, and ASICs in ischemia and inflammatory pain, and have been implicated in memory and learning."

2. 2nd paragraph on p.4, line 4, "Both form a homotrimer...". The high resolution structures obtained so far are from ASIC1a homotrimers; however ENaC was expressed as heterotrimer of alpha, beta and gamma subunits. This should be corrected. As a reference for the ENaC structure, Noreng et al., 2020 may be more appropriate than the 2018 article, since it is more complete.

Thank you for pointing that out, we have included the homomeric and heteromeric structures in the text. Initially, we used 2018 because we referred to the first ENaC structure but we have added Noreng et al., 2020 and changed "Both form a homotrimer..." to "Both form a trimer..." (line 64).

3. A little bit further down, "The structure of each protomer..", the central, beta-strand-rich scaffold is formed by the palm, not the beta-ball.

This part has been re-written to clarify according to the reviewer's comment (line 67).

4. Pore, gate, selectivity filter (bottom of p.4 and top of p.5). You should be careful to not mix up "gate" and "selectivity filter". The gate at the level of Asp433 is not put into question by the newer studies. It was never stated, contrary to what you mention, that Asp433 contributed to ion selectivity. The "GAS-belt" is mainly involved in ion selectivity. The structural article by Yoder and a functional study (doi:10.7554/eLife.24630) questioned somewhat this role, identifying acidic residues further down the pore contributing to the selectivity. A recent study that you cite (Chen et al., 2022) suggested then that the effect of these lower acidic residues on ion selectivity is indirect, via the GAS-belt. It would be good to rewrite this part on gating and ion selectivity. Regarding the last sentence in this paragraph, the Na, K or Ca selectivity, I was not aware of K-selective members. Are these really K-selective, or just non-selective cation channels?

Thank you, we have re-written this section on gate and selectivity filter (line 80) and we have re-phrased the K-selectivity statement (line 94).

5. Fig. 1. This is a nice and clear figure. The calcium ions are shown as "2+". Could you mark them as "Ca2+" or at least indicate in the legend that "2+" means Ca ions? You mention in the legend also that the N- and C-termini are short. In ENaC they are 100 residues long or longer. You should relativize this statement.

Apologies, we have submitted a new version of **Figure 1**, where we have corrected “2+” to “Ca²⁺”, and we have edited the statement to “The N- and the C-terminal regions are relatively short in many family members”

6. Fig. 2. For the trace in figure 2B, can you indicate which Ca indicator was used, or show a reference?

We have included in the legend that the ratiometric indicator Cameleon was used and included a reference to the study to which the example trace contributed (Walker & Schafer 2020).

7. p9, last paragraph, "Thus, in the TRNs,..." Could you better explain what the basis of the mentioned hypothesis is. The sentence following this one would rather suggest the opposite of the hypothesis.

Apologies, in the interest of brevity, we did not explain very clearly! We have edited, referring the reader back to the genetic evidence for the roles of MEC-4 and MEC-10 in the previous paragraph, to link these observations. We have also added more detail to our description of the heterologous expression experiments to make clearer that the lack of response for MEC-4 alone does not necessarily reflect a lack of mechanosensitivity, rather, that a different experimental approach may be required to demonstrate it (**line 127**).

8. p.10, bottom and p11 top, the tethering model. In a study by M. Goodman of 2002 (doi 10.1038/4151039a) it was shown that co-expression of MEC-4 and MEC-10 with MEC-2 was sufficient to express ion currents in *Xenopus* oocytes. So it seems that tethering on both sides is not necessary. Please discuss this evidence in the text.

Goodman et al. used a dominant allele of *mec-4*, since expression of wild-type *mec-4* did not give currents, even with other potential subunits. The currents in that paper are not mechanically induced, but are from this mutant form of the channel that does not need to be mechanically gated, so are not directly relevant to the tethering model.

9. Actin and ENaC (bottom of p11). The paper by Berdiev is based on ENaC measurements in an artificial system. It seems surprising that actin should affect the single-channel conductance. Could you provide more information with this statement, or take it out?

We have added a more appropriate reference (**line 191**).

10. p12, "ASIC function in noxious mechanosensation...". Several aspects discussed in this paragraph are not really noxious functions. Could you adapt the title? You mention a role of ASIC3 in stretch response. The indicated reference is a method paper, and I did not find any mention of ASIC3. Could you double-check it? An important article on the unsuccessful cellular measurement of an involvement of ASICs in mechanosensation is the article by Drew and Wood (PMID: 14990679). From studies with KO mice there is also evidence for a mechanosensory role of ASICs in the gastrointestinal tract. It might be worth to mention it.

We have amended the title, and added the correct reference for the stretch response role. We agree that the Drew reference should be included, but have also considered the potential reasons for their negative result, countering with the observations of Lin et al. Thank you for the suggestion of also including gastrointestinal pain. In accommodating these improvements, we have also used the opportunity to emphasise the link between inflammation and mechanosensory roles (line 207).

11. p13, " A model for understanding degeneration". In the first lines, some mutations of Deg/ENaC channels in *C. elegans* are mentioned that were used for genetic screens. It would be good to mention that these mutations allow uncontrolled Na entry leading to cell death. This information would help for the understanding of what follows. The explanation is given at the bottom of the paragraph. It would be good to move it up. This would make the text more logical.

Thank you, we agree, we have moved this to the top of the paragraph (line 250)

12. bottom line of p.13, "...with the rat ASIC3 being the first gene cloned...". This is not correct. ASIC1a and ASIC3 were published in the same year, and since the identification of ASIC1a was published before that of ASIC3, it was published in nature. What was new in these two papers was that the authors had found out that ASICs are activated by low pH. In the years before, several ASICs had been cloned by the laboratories of Michel Lazdunski, Michael Welsh and David Corey, and they had called them "brain Na channel" or similar, and had at this time not found any way of activating them, except by the degenerin mutation. For this reason, it is not correct to state that ASIC3 was the first cloned ASIC.

Thank you very much for this very helpful correction. We have revisited this statement about the first channels cloned and the first time that acid-activation was shown and included all the suggested literature (line 294).

13. p. 14, different groups of acid-regulated channels. For the "acid- activated" group, it would be useful to distinguish channels that are indeed activated by acid (i.e. from completely closed to open), as the ASICs, and channels whose activity is modulated by pH. To this second group would for example belong the human ENaC, whose basal activity can be changed by +/- 30% depending on the pH.

Thank you, that is a great suggestion. We further divided the acid-activated group into two subgroups based on this suggestion (line 303).

The BLINAC is in this context named ASIC5. Since this channel is not closer to ASIC than it is to other ENaC/Deg members, why not calling it "BASIC", as recently proposed?

Thank you for this suggestion, we included all names now (line 321).

14. p.4, bottom, "In contrast to its derivative benzamil...". amiloride and benzamil are closely related and bind to the same site in ENaC/Deg channels. The observed difference in

voltage dependence between the two compounds may be due to different conditions, different channels, etc. If you want to mention this difference, you should probably better describe the conditions of studies investigating amiloride and benzamil.

This is based on findings from Fechner et al. However, as discussing the experimental conditions in more detail would detract from the aim of this section, which was to provide a very brief overview, we have removed this statement (line 372).

15. bottom of p.15, "Likewise, lactate...", the reference is not correct, it should be PMID: 11528414 DOI: 10.1038/nn0901-869.

We are confused by this comment. The suggested reference is the same reference that we have in the text.

16. p16, lower paragraph, discussion of toxins. Not all the statements in this paragraph are completely correct. Mambalgin inhibits ASIC1a but also ASIC1b. Psalmotoxin inhibits according to most of the studies only ASIC1a homotrimers, but not ASIC1a-containing heterotrimers. It is correct that APETx2 inhibits only ASIC3 of the ASICs, however, it is also an inhibitor of some voltage-gated Na channels (Peigneur et al., PMID: 22972919 and Blanchard et al., PMID: 21943094). Therefore, the usefulness of APETx2 in animal studies is somewhat limited. I suggest to re-formulate some of the statements.

Thank you, we have included ASIC1b Mambalgin-1 inhibition. Regarding psalmotoxin inhibition, we have added 2 references to support our statement that it inhibits heteromers. We also agree that it is interesting to briefly mention the wider promiscuity of APETx2 (line 430).

17. p20, second paragraph, "Likewise, in vivo...". There is not sufficient information shown to really understand the statement. It would be good to add somewhat more information, e.g. indicating in which type of neuron this activity was measured, what was the basis for the conclusion that the role is presynaptic.

We have added that this observation relates to the *Drosophila* mdIV neurons. The hypothesis of a presynaptic role is based on the late observation that Ppk1 localises presynaptically (Orr et al., 2017). We have included the information in the text (line 520).

18. p21, top paragraph, "Vertebrate ASICs are also expressed in glia...". This sentence gives a wrong impression of the current situation. In fact, there are many papers showing high ASIC currents in neurons and only two papers showing ASIC activity in glial cells of vertebrates, and only one of these papers provided current traces, which showed that the currents in the glial cells are quite small. Many articles addressing learning have clearly shown that the role of ASICs was in the neurons. Therefore, for learning in vertebrates, the current task is not to disentangle neuronal from glial functions, but rather to find whether glial ASICs may also have roles in learning.

We have added “gene” to make clear that by expression we mean gene expression, not currents. For brevity, we have not included all of the references showing glial expression, but instead have referred to Hall & Ben-Shahar and Cegielski et al. (who each list multiple examples). We have also tried to justify our reason for speculating on glial roles in plasticity more clearly, but in recognition of the reviewer’s point, we have also reworded to say that ASIC roles in learning may include a glial element. Finally, we felt that it was important to emphasise the challenge of distinguishing between neuronal and glial sites of function, using this as an example of the difference between invertebrate and vertebrate models, in terms of the ease with which such a question can be addressed (line 543).

Referee #2:

This is an interesting review that summarizes diverse findings on DEG/ENaC ion channels. While the review will be interesting to many readers interested in ion channels, it would still profit from a stronger focus and from a more critical summary of sometimes conflicting published data.

General comments:

1) This review does not have a clear focus. It tries to cover many diverse and unrelated aspects of DEG/ENaC channels. I think it would profit from a clearer focus on a subset of specific aspects, as for example mechanosensitivity or proton sensitivity of DEG/ENaCs.

Thank you for this suggestion. The aim of the review was to cover the diversity of this channel family, with a focus on bringing together genetics and physiology, and how both in combination can be used to understand unique and common functions. We have added summary sentences and more explanation in the text in order to make the focus of the review more obvious, and to illustrate where crosstalk between physiology and genetics, and vertebrates and invertebrates has been, or could be, impactful. For example, when discussing stomatins and paroxonases (line 238, 243); demonstrating mechanosensitivity of MEC-4/MEC-10 (line 130); the importance of relating channel properties and genetics/behaviour, in relation to the acid-sensing members (line 339) and synaptic function (line 554). The journal guidelines do not allow us to use the Summary to outline what we will discuss, so we have slightly reorganised the introductory section to allow us to expand (line 49) to try to state our aims more clearly and to reiterate the importance of interaction between sub-fields.

2) I had the impression that the authors strive to find overarching and common functions for DEG/ENaCs from different species. But it appears that DEG/ENaCs have surprisingly diverse and often still enigmatic functions. Therefore, highlighting common functions and properties across many different species ignores conflicting data and over-simplifies the state-of-knowledge. Moreover, controversial opinions in the field are ignored. But I think

the reader would profit from a more balanced and more critical review of different opinions and different findings. The authors paint a picture that is, in my opinion, too black-and-white.

Yes indeed, our aim was to identify commonalities and overarching themes across species, but we completely agree that many questions remain regarding the degree to which functions are actually conserved. We have expanded the discussion to make this point and also to point out some of the current gaps in our knowledge – the incompleteness of data, with many family members uncharacterised, and the difficulty of comparing when the emphasis has been on different technical approaches in different organisms (line 563). We have also tried to take a more critical, balanced stance. For example, in relation to neuropeptides (line 419), the role of ASICs in mechanosensation (line 205), mechanosensitivity of *mec-4* (line 132) and acid sensitivity and the questions of evolution and commonality of mechanism (line 333).

3) Often many findings on DEG/ENaC functions are simply summarized without providing context. Sometimes this suggests conclusions that are misleading. I provide some examples below in my specific comments.

4) "Cross-phylum evidence for ASICs as proton-sensors." In this section, the authors give the over-simplified impression that ASICs in many different phyla act as proton-sensors. This is somewhat simplistic and misleading. The authors should consider that it will be hard to find any ion channel that is NOT in some way influenced by protons. Inhibition by acid, for example, may have many reasons, for example a simple block of the ion pore by permeating protons. This does not prove an evolutionary conserved mechanism.

We agree that the term acid sensors may be over-interpreting the evidence, so have edited the title of this section and added a statement at the end raising the question of conservation of function/evolution/extent to which the property is functionally relevant (line 333). We also agree that almost all ion channels are likely to show some degree of pH sensitivity. However, the evidence presented certainly does not amount to a simple blocking of the pore at high proton concentration; an array of channels (e.g. Martí-Solans et al. 2022, Kaulich et al 2022a and 2022b) from different invertebrate lineages, exhibit activation at high pH, or activation over a narrow pH range with inhibition at both high and low pH. We have also substantiated our channel property observations with evidence for roles for these proton sensing properties *in vivo* (Kaulich et al 2022a).

5) I found the section entitled "Many ligands differentially modulate DEG/ENaC channel activity" particularly weak. The author define a ligand in a very broad sense. From pharmacological inhibitors to cations, serotonin and lactate. The mode of action of all these "ligands" is very different. Lactate actually does not bind to ASICs but simply chelates Ca²⁺. After the next section on "Neuropeptides and toxins", the authors conclude "that DEG/ENaC are modulated by a diverse array of ligands". Are amiloride and NSAIDs really ligands modulating DEG/ENaCs? And why should these drugs or toxins have a "role in vivo"? This part is more confusing than enlightening.

Yes, we defined it in a very broad sense, but we see the point of the reviewer, therefore, we changed the heading to "Diverse array of modulators of DEG/ENaC channels" (line 344), restructured the section and ensured that we better explain the mode of action.

6) Personally, I found also the concluding remarks too enthusiastic and uncritical. How likely is it really that DEG-1 and PPK29 and PPK11 have so divergent functions? Isn't it also possible (and perhaps more likely) that we still have not understood all their roles well enough?

Thank you, we agree that this is a very important point. We have expanded the discussion to make this point, using the examples of pH sensitivity (are they in fact all pH sensitive, as so many channel types are?) and mechanosensation, as well as discussing the challenges presented by the bias for specific experimental approaches in different species (line 563).

Specific comments:

1) For the lay reader, please give a brief explanation of the terms "forward genetic screen" and "reverse genetic screen".

Of course, we have now added a brief explanation of these terms (line 33).

2) Page 2: "Protons bind ASICs in this acidic pocket". The essential proton binding sites of ASICs are still unknown. Probably there are many of them and some outside the acidic pocket. This complex view is not really reflected here.

Yes, we agree, we have rephrased this statement and we have included a reference in the discussion of binding sites for protons (Rook et al., 2021) (line 77).

3) I think for some of the figures, it would be fair to mention the sources that they are based on.

We sincerely apologise, this must have been deleted during our editing process, we think this comment refers to fig. 1a in which the structure was based on (Gruender and Chen, 2010). We have put it back into the legend. Sincere apologies for that, and thank you very much for noticing! The trace in figure 2B was part of the data set of Walker & Schafer 2020, so we have also referenced that.

4) Page 9: "Pickpockets ... are specifically required for response to mechanical stimuli." What does this mean exactly? Are they mechanosensitive ion channels? Is the evidence unequivocal? What is their specific function for responses to mechanical stimuli?

Apologies if this was not clear. The intended meaning of this sentence was simply that they are specifically required for the (behavioural) response (just described) to mechanical, as opposed to thermal or chemical, noxious stimuli. We have edited to make this clearer (line 146). At the end of the paragraph, we already stated that demonstrating mechanosensitivity had proven challenging; we have added to this statement to emphasise that this means that

we do not know whether their behavioural role reflects a direct role in mechanotransduction.

5) Page 10: "Their [ENaCs] wide-ranging functions are reflected in the multi-system defects seen in hereditary diseases (Liddle syndrome, type I PHA and CF-like diseases)". The defects in these diseases are all related to Na⁺ reabsorption and do NOT reflect wide-ranging functions.

We have adjusted the wording (line 180). We did not intend to mean wide-ranging channel functions; rather that they are widely expressed and thus are important in the function of multiple tissues/organs.

6) Page 10: What are "non-canonical ENaCs"?

ENaC homomers and heteromers consisting of subunit compositions different to the "canonical" $\alpha\beta\gamma$ subunits (Baldwin et al., 2020). We included an explanation (line 186).

7) Page 11: Why does cholesterol-dependence of proper localization of DEG/ENaC channels indicate a role in determining the lipid environment of the channel? Isn't it the opposite?

Apologies, this was not very clear. We meant that the evidence is consistent with a role for MEC-2 in determining the lipid environment, i.e. recruiting/maintaining cholesterol in the channel complex. We have edited to explain this better (line 235).

8) Page 11: What is the relevance of modulation of ASIC2 and ASIC3 by stomatin and STOML3?

We have edited the wording to better explain that this suggests functional conservation, reiterating the point later in the paragraph, in relation to MEC-6/paroxonases and adding that this illustrates the utility of crosstalk between scientists working on the different family members and using different technical approaches (line 238)

9) Page 12: whether the unregulated Na⁺ and Ca²⁺ entry of ASIC1a leads to neuronal death in ischemia is still controversial. In contrast to Mec/DEG genes with constitutively activating mutations, ASICs desensitize, limiting the ion flow at prolonged activation.

Absolutely, we have included this in the manuscript and added other references to put it in context (including different results and the role of potential other channels and kinases interacting with ASICs) (i.e. Vukicevic & Kellenberger, 2004; Petroff et al., 2008; Mazzone et al., 2017, William et al., 2020) (line 246).

10) ASIC3 was NOT the first ASIC gene to be cloned.

We apologise, we have now included the first ASICs cloned and the changed the wording to the first time that acid-sensitivity for these channels was shown (line 294).

11) Page 16: The hypotheses on ASICs as sensors of sour and salt taste are quite old and have not been confirmed.

We absolutely agree, and have adjusted this section to try to reflect this better (lines 455, 477).

12) To give a positive example: I found the discussion of the role of PPKs in courtship behaviour interesting and helpful. For example, figure3 nicely illustrates two alternative models. I would have liked to see more of such balanced view that may informed future studies.

Thank you! We have tried to present a more balanced view, as outlined in our response to point 2.

13) Page 19: C. elegans ASIC-1 is unrelated to (not the species ortholog of) mammalian ASIC1. This information is lacking here and the discussion of ASIC-1 is therefore somewhat misleading.

Thank you, we agree that it is important to point out that the name could be misleading, we have made this clear (line 509). We have also added a note to remind the reader that, as we recently showed, ASIC-1 is acid-activated, fitting with the hypothesis of Voglis and Tavernarakis (line 516).

REQUIRED ITEMS:

-Please include a full title page as part of your article (Word) file (containing title, authors, affiliations, corresponding author name and contact details, keywords, and running title).

-Please include a legend to accompany your abstract figure.

-Please upload separate high quality figure files via the submission form.

-It is the authors' responsibility to obtain any necessary permissions to reproduce previously published material https://jp.msubmit.net/cgi-bin/main.plex?form_type=display_requirements#use

END OF COMMENTS

Dear Dr Walker,

Re: JP-TR-2022-283335R1 "The diverse functions of the DEG/ENaC family: linking genetic and physiological insights" by Eva Kaulich, Laura J Grundy, William R Schafer, and Denise S Walker

Thank you for submitting your Topical Review to The Journal of Physiology. It has been assessed by a Reviewing Editor and by 2 expert referees and I am pleased to tell you that it is considered to be acceptable for publication following satisfactory revision.

The reports are copied at the end of this email. Please address all of the points and incorporate all requested revisions, or explain in your Response to Referees why a change has not been made.

NEW POLICY: In order to improve the transparency of its peer review process The Journal of Physiology publishes online as supporting information the peer review history of all articles accepted for publication. Readers will have access to decision letters, including all Editors' comments and referee reports, for each version of the manuscript and any author responses to peer review comments. Referees can decide whether or not they wish to be named on the peer review history document.

I hope you will find the comments helpful and have no difficulty in revising your manuscript within 4 weeks.

Your revised manuscript should be submitted online using the links in Author Tasks Link Not Available. This link is to the Corresponding Author's own account, if this will cause any problems when submitting the revised version please contact us.

You should upload:

- A Word file of the complete text (including any Tables);
- An Abstract Figure, (with accompanying Legend in the article file)
- Each figure as a separate, high quality, file;
- A full Response to Referees;
- A copy of the manuscript with the changes highlighted.
- Author profile. A short biography (no more than 100 words for one author or 150 words in total for two authors) and a portrait photograph of the two leading authors on the paper. These should be uploaded, clearly labelled, with the manuscript submission. Any standard image format for the photograph is acceptable, but the resolution should be at least 300 dpi and preferably more.

- A 'Cover Art' file for consideration as the Issue's cover image;
- Appropriate Supporting Information (Video, audio or data set https://jp.msubmit.net/cgi-bin/main.plex?form_type=display_requirements#supp).

To create your 'Response to Referees' copy all the reports, including any comments from the Senior and Reviewing Editors into a Word, or similar, file and respond to each point in colour or CAPITALS. Upload this when you submit your revision.

I look forward to receiving your revised submission.

Yours sincerely,

Professor Laura Bennet
Senior Editor
The Journal of Physiology
<https://jp.msubmit.net>
<http://jp.physoc.org>
The Physiological Society
Hodgkin Huxley House
30 Farringdon Lane
London, EC1R 3AW
UK
<http://www.physoc.org>
<http://journals.physoc.org>

EDITOR COMMENTS

Reviewing Editor:

We thank the authors for the making revisions to the manuscript "The diverse functions of the DEG/ENaC family: linking genetic and physiological insights" in response to comments raised by the referees. Whilst the previous comments have largely been answered in the revision, both referees have noted some remaining minor points which need to be addressed.

REFEREE COMMENTS

Referee #1:

The authors have addressed the points that I have raised.

There are just two points that the authors should still look at:

1. On line 78, there is a sentence: "Based on these structures, it was long thought that ASICs to have a primary gate (upper gate) in the channel, contributed by Asp433 (numbering in chicken ASIC1a) from each monomer (figure 1E)." The way this is formulated, it gives the impression that something is wrong with this conclusion. I suggest to change this statement somewhat, as e.g. "These structures identified, together with functional studies, the existence of an upper gate in ASIC1, contributed by Asp433 (numbering in chicken ASIC1a) from each monomer (figure 1E)."

2. This concerns the statement about the ion selectivity (line 93). I have the impression that the authors mix up "permeable" with "selective". Based on my knowledge about this family, most of the ENaC/DEG members are Na-selective; some are in addition permeable to K and Ca. I suggest to formulate this sentence differently, as for example: "...; indeed, most are selective for Na⁺ and some of them are also permeable to K⁺ and/or Ca²⁺."

Referee #2:

I appreciate the efforts of the authors to substantially improve their review. I have a few remaining minor comments, part of which arise from the fact that the authors do still not always sufficiently differentiate between different DEG/ENaCs. I hope they find my comments useful to further improve their review.

Minor comments:

1) Page 3/4, Section on "Structural insights": it is not always clear whether the authors refer to ASICs and ENaC or only to one of these subgroups. For example (lines 71/72), "The interface between the thumb, b-ball and the finger region encompasses the primary acidic pocket..." Does ENaC also have an acidic pocket? On the other hand (line 89-91), "Both motifs are implicated in gating and ion selectivity" All the references given refer to ENaC and not to ASICs. Please be more specific.

2) Line 259: "...it has been proposed that acidosis is caused indirectly...". I think the authors really mean "...it has been proposed that cell death is caused indirectly..."

3) Line 261: I think rather than citing the comment by William et al, it would be more appropriate to cite the original articles by the Xu group (PMID: 26523449 and PMID: 31980622).

4) Page 10, section on "Acid-sensing members": as "acidosis" usually refers to acidity of the blood plasma, I suggest to change the term "acidosis" by "acidic pH". For example, lines 279 and 281.

5) Lines 284/285: "Insights into acid-sensitive DEG/ENaCs are largely based on studies of heterologously expressed channels". Although the term "largely" leaves room for interpretation, this statement gives a wrong impression and does not do justice to the numerous high-quality studies that have been done with animals (mainly knock-out mice). Some of these studies are indeed discussed later (top of page 17).

Similar concern for lines 417/418.

6) Lines 291-316: The authors state (line 293) that "Acid-sensitive DEG/ENaCs classify into three main groups." Then (on line 293) "the acid-activated group..." That seems to be the first of three groups. On line 309 it reads "The second group, the acid-inhibited channels,..." I was left wondering what the third main group would be. Or did I miss something?

7) Lines 306-307: "Both subgroups generate large proton-activated inward currents, with channel-specific half-activation pH varying from around 6.5 to 4.3" These two subgroups also encompass ENaC. Later (lines 318/319), however, it reads "human abg ENaC channels' pH range (pH 8.5-6) fits with..." This range seems inconsistent with the previously mentioned pH values. Please be more specific. What is the pH sensitivity of ENaCs and what is it of ASICs?

8) Lines 360/361: "Structural and mutational studies have revealed that it [amiloride] binds within the open channel pore, on the extracellular acidic pocket". I understand this sentence that the channel pore is on the acidic pocket, which is of course not the case. Or do the authors mean "channel pore and acidic pocket"? It would be helpful to clearly mention that the amiloride binding site in the pore accounts for inhibition and the other site on the acidic pocket might account for the paradoxical stimulation of some DEG/ENaCs by amiloride.

9) Line 372: "...illustrating functional conservation..." I do not believe that a conserved binding site for a synthetic drug can allow conclusions about conserved functions of ion channels.

10) Page 15, section on "DEG/ENaCs and taste". The evidence for different DEG/ENaCs from different species is somewhat mixed here, which is sometimes confusing. For example, lines 435/436, "...since salt-taste is blocked by amiloride, that they are salt-taste receptors." To which species does the block by amiloride of salt-taste refer and does "they" in the last part of this sentence refer to "ASICs" or to "DEG/ENaCs". Please be more specific.

11) Finally, I would like to propose a table summarizing the functions of different PPKs. Because there are so many and because the numbers of PPKs do not provide any clue to their function, this would be particularly helpful. Such a table is of course not mandatory.

END OF COMMENTS

1st Confidential Review

22-Sep-2022

13th October 2022

Dear Editors and reviewers,

Re: JP-TR-2022-283335R1 "The diverse functions of the DEG/ENaC family: linking genetic and physiological insights".

Thank you for your very valuable feedback on our revised manuscript. Please find below our individual responses to each of the comments, outlining in purple how we have addressed them. In addition, we have updated the references to correct the citation for Kaulich *et al.* 2022a and to update Kaulich *et al.* 2022b from bioRxiv to the published J. Physiol. citation.

Best wishes,

Denise Walker.

EDITOR COMMENTS

Reviewing Editor:

We thank the authors for the making revisions to the manuscript "The diverse functions of the DEG/ENaC family: linking genetic and physiological insights" in response to comments raised by the referees. Whilst the previous comments have largely been answered in the revision, both referees have noted some remaining minor points which need to be addressed.

Referee #1:

The authors have addressed the points that I have raised.

There are just two points that the authors should still look at:

1. On line 78, there is a sentence: "Based on these structures, it was long thought that ASICs to have a primary gate (upper gate) in the channel, contributed by Asp433 (numbering in chicken ASIC1a) from each monomer (figure 1E)." The way this is formulated, it gives the impression that something is wrong with this conclusion. I suggest to change this statement somewhat, as e.g. "These structures identified, together with functional studies, the existence of an upper gate in ASIC1, contributed by Asp433 (numbering in chicken ASIC1a) from each monomer (figure 1E)."

Thank you for this suggestion. We have edited as suggested (line 76).

2. This concerns the statement about the ion selectivity (line 93). I have the impression that the authors mix up "permeable" with "selective". Based on my knowledge about this family, most of the ENaC/DEG

members are Na-selective; some are in addition permeable to K and Ca. I suggest to formulate this sentence differently, as for example: "...; indeed, most are selective for Na⁺ and some of them are also permeable to K⁺ and/or Ca²⁺."

We agree that this sentence is unclear! We have edited to clarify as suggested, as well as adding an additional reference (Kaulich et al) to further support the statement that not all are preferentially Na-selective (line 93).

Referee #2:

I appreciate the efforts of the authors to substantially improve their review. I have a few remaining minor comments, part of which arise from the fact that the authors do still not always sufficiently differentiate between different DEG/ENaCs. I hope they find my comments useful to further improve their review.

Minor comments:

1) Page 3/4, Section on "Structural insights": it is not always clear whether the authors refer to ASICs and ENaC or only to one of these subgroups. For example (lines 71/72), "The interface between the thumb, b-ball and the finger region encompasses the primary acidic pocket..." Does ENaC also have an acidic pocket? On the other hand (line 89-91), "Both motifs are implicated in gating and ion selectivity" All the references given refer to ENaC and not to ASICs. Please be more specific.

We have edited this section to make clear when we are talking about ASIC or ENaC, adding the references again where appropriate, as well as adding a brief description of the key differences (lines 67-98).

2) Line 259: "...it has been proposed that acidosis is caused indirectly...". I think the authors really mean "...it has been proposed that cell death is caused indirectly..."

Apologies – acidosis should have read acidotoxicity. We have corrected it (line 260).

3) Line 261: I think rather than citing the comment by William et al, it would be more appropriate to cite the original articles by the Xu group (PMID: 26523449 and PMID: 31980622).

Absolutely. Apologies for this oversight. We have inserted the correct references (line 2).

4) Page 10, section on "Acid-sensing members": as "acidosis" usually refers to acidity of the blood plasma, I suggest to change the term "acidosis" by "acidic pH". For example, lines 279 and 281.

Thank you for pointing out this potential confusion. We have edited those two to "acidic pH" (lines 280, 282) and ensured that elsewhere we have clarified where we are referring to local acidosis (as the sources to which we refer have done). For example, we compare local lactic acidosis to systemic acidosis (line 357).

5) Lines 284/285: "Insights into acid-sensitive DEG/ENaCs are largely based on studies of heterologously expressed channels". Although the term "largely" leaves room for interpretation, this statement gives a wrong impression and does not do justice to the numerous high-quality studies that have been done with animals (mainly knock-out mice). Some of these studies are indeed discussed later (top of page 17).

We have edited "largely based" to "founded", which we hope better reflects our intended meaning (line 286).

Similar concern for lines 417/418.

We hope that our clarification of our meaning addresses this concern (line 421).

6) Lines 291-316: The authors state (line 293) that "Acid-sensitive DEG/ENaCs classify into three main groups." Then (on line 293) "the acid-activated group..." That seems to be the first of three groups. On line 309 it reads "The second group, the acid-inhibited channels,..." I was left wondering what the third main group would be. Or did I miss something?

We have added "the third group" to make this clearer (line 315)

7) Lines 306-307: "Both subgroups generate large proton-activated inward currents, with channel-specific half-activation pH varying from around 6.5 to 4.3" These two subgroups also encompass ENaC. Later (lines 318/319), however, it reads "human abg ENaC channels' pH range (pH 8.5-6) fits with..." This range seems inconsistent with the previously mentioned pH values. Please be more specific. What is the pH sensitivity of ENaCs and what is it of ASICs?

Apologies. What we meant was that the channel has maximal activity at pH6/minimal activity at pH 8.5 We have edited to clarify (line 319).

8) Lines 360/361: "Structural and mutational studies have revealed that it [amiloride] binds within the open channel pore, on the extracellular acidic pocket". I understand this sentence that the channel pore is on the acidic pocket, which is of course not the case. Or do the authors mean "channel pore and acidic pocket"? It would be helpful to clearly mention that the amiloride binding site in the pore accounts for inhibition and the other site on the acidic pocket might account for the paradoxical stimulation of some DEG/ENaCs by amiloride.

Thank you for spotting this error, this sentence was indeed incomplete; we have filled in the gaps (line 362).

9) Line 372: "...illustrating functional conservation..." I do not believe that a conserved binding site for a synthetic drug can allow conclusions about conserved functions of ion channels.

Our point was simply that conservation of a binding site for these specific drugs indicates conservation of structure and thus functional relevance of the invertebrate counterparts. We have edited to "structural" (line 374).

10) Page 15, section on "DEG/ENaCs and taste". The evidence for different DEG/ENaCs from different species is somewhat mixed here, which is sometimes confusing. For example, lines 435/436, "...since salt-taste is blocked by amiloride, that they are salt-taste receptors." To which species does the block by amiloride of salt-taste refer and does "they" in the last part of this sentence refer to "ASICs" or to "DEG/ENaCs". Please be more specific.

Thank you for this feedback. We agree that the mixed species evidence made this section confusing. We have rearranged the paragraph to first deal with ASICs and then bring in the invertebrate members (line 430).

11) Finally, I would like to propose a table summarizing the functions of different PPKs. Because there are so many and because the numbers of PPKs do not provide any clue to their function, this would be particularly helpful. Such a table is of course not mandatory.

We absolutely agree that this would be a very useful resource. Likewise, the vast majority of the *C. elegans* names do not reveal any clue as to their function, so it would seem inappropriate to do this for one without the other (or indeed members from other species). Such a table would need to be exhaustive and would therefore add significantly to the length of the review and the number of references required, so would perhaps be better placed in a review dedicated to them.

Dear Dr Walker,

Re: JP-TR-2022-283335R2 "The diverse functions of the DEG/ENaC family: linking genetic and physiological insights" by Eva Kaulich, Laura J Grundy, William R Schafer, and Denise S Walker

I am pleased to tell you that your Topical Review article has been accepted for publication in The Journal of Physiology, subject to any modifications to the text that may be required by the Journal Office to conform to House rules.

NEW POLICY: In order to improve the transparency of its peer review process The Journal of Physiology publishes online as supporting information the peer review history of all articles accepted for publication. Readers will have access to decision letters, including all Editors' comments and referee reports, for each version of the manuscript and any author responses to peer review comments. Referees can decide whether or not they wish to be named on the peer review history document.

The last Word version of the paper submitted will be used by the Production Editors to prepare your proof. When this is ready you will receive an email containing a link to Wiley's Online Proofing System. The proof should be checked and corrected as quickly as possible.

All queries at proof stage should be sent to tjp@wiley.com

The accepted version of the manuscript will be published online, prior to copy editing in the Accepted Articles section.

Are you on Twitter? Once your paper is online, why not share your achievement with your followers. Please tag The Journal (@jphysiol) in any tweets and we will share your accepted paper with our 22,000+ followers!

Yours sincerely,

Professor Laura Bennet
Senior Editor
The Journal of Physiology
<https://jp.msubmit.net>
<http://jp.physoc.org>
The Physiological Society
Hodgkin Huxley House
30 Farringdon Lane
London, EC1R 3AW
UK
<http://www.physoc.org>
<http://journals.physoc.org>

*** IMPORTANT NOTICE ABOUT OPEN ACCESS ***

To assist authors whose funding agencies mandate public access to published research findings sooner than 12 months after publication The Journal of Physiology allows authors to pay an open access (OA) fee to have their papers made freely available immediately on publication.

You will receive an email from Wiley with details on how to register or log-in to Wiley Authors Services where you will be able to place an OnlineOpen order.

You can check if your funder or institution has a Wiley Open Access Account here <https://authorservices.wiley.com/author-resources/Journal-Authors/licensing-and-open-access/open-access/author-compliance-tool.html>

Your article will be made Open Access upon publication, or as soon as payment is received.

If you wish to put your paper on an OA website such as PMC or UKPMC or your institutional repository within 12 months of publication you must pay the open access fee, which covers the cost of publication.

OnlineOpen articles are deposited in PubMed Central (PMC) and PMC mirror sites. Authors of OnlineOpen articles are permitted to post the final, published PDF of their article on a website, institutional repository, or other free public server, immediately on publication.

Note to NIH-funded authors: The Journal of Physiology is published on PMC 12 months after publication, NIH-funded authors DO NOT NEED to pay to publish and DO NOT NEED to post their accepted papers on PMC.

EDITOR COMMENTS

Reviewing Editor:

We thank the authors for responding to the minor comments raised by the reviewers, these have now been adequately addressed.

2nd Confidential Review

13-Oct-2022